# Structural analyses uncover protease-adhesin interactions and c-di-GMP receptor regulation in sulfate-reducing bacteria

Maria E. Font[1], Amruta A. Karbelkar[2], Justin D. Lormand[1], Sofia Mortensen [1], María J. García-García [1], George A. O'Toole[2] & Holger Sondermann [1,3] ✉

*Desulfovibrio vulgaris* is a sulfate-reducing organism with biofim-forming capacity relevant for bioremediation and microbe-induced corrosion. Biofilm formation of *D. vulgaris* depends on two large adhesins that are regulated by proteins encoded in the Dvh operon, which resembles the gammaproteobacterial Lap system in composition but differs in the sequence and domain organization of its regulatory proteins, DvhG and DvhD. We show that DvhG is a calcium-dependent protease that targets the periplasmic domains of both adhesins via extensive interactions. Additionally, structures of DvhD establish this HD-GYP domain-containing protein as a c-di-GMP-dependent switch with a periplasmic dCache domain. Our data support a model in which DvhD controls DvhG activity through a c-di-GMP-dependent mechanism that is molecularly distinct, but functionally analogous to LapD. Together, our results reveal how conserved regulatory logic can be implemented through distinct molecular architectures, highlighting the evolutionary flexibility of c-di-GMP signaling networks in controlling surface attachment across diverse bacterial lineages.

Bacterial biofilms, multicellular communities of microorganisms embedded in a self-produced extracellular matrix, are ubiquitous in nature and found in environments from soil to aquatic habitats and human tissues[1]. The biofilm matrix allows bacteria to colonize surfaces and forms a protective layer that confers resistance to antibiotics, host immune defenses, and environmental stresses. Biofilms play important roles in medicine, industry, and ecology, with both beneficial and detrimental effects. Therefore, understanding biofilm formation and dispersal mechanisms is crucial for developing strategies to control chronic infections, promote beneficial microbial colonization, and optimize industrial processes.

A crucial step in biofilm formation is the deployment of large, cell surface-localized adhesin proteins, which mediate cell-cell, cell-matrix, and cell-substratum interactions[2-11]. Gammaproteobacteria deploy large adhesins under the control of the intracellular second messenger c-di-GMP[12]. A particular class of c-di-GMP-controlled adhesins, exemplified by *Pseudomonas fluorescens* LapA, is

transported and inserted into the outer membrane via a distinct type 1 secretion system (T1SS), in which the adhesin transverses and anchors to the outer membrane TolC-like transmembrane channel LapE to support cell attachment (Fig. 1a, left)[13]. Alternatively, the opportunistic pathogen *P. aeruginosa* expresses a distinct type Vb secretion system, where the adhesin is translocated and anchored in the outer membrane via a syntenic autotransporter[14]. In both cases, adhesins contain a retention module that resides in the periplasm and can be proteolytically cleaved by the periplasmic protease LapG in a c-di-GMP-dependent fashion (Fig. 1a, right). Specifically, an inner membrane receptor, LapD, binds c-di-GMP in the cytosol and LapG at its periplasmic domain[15-17]. In the LapD•c-di-GMP•LapG complex, LapG cannot proteolyze the target sequence within the adhesins, ensuring they remain anchored in the outer membrane. When cellular c-di-GMP levels drop, c-di-GMP dissociates from LapD and releases LapG, which is then able to cleave the retention module from the adhesin[17-19]. Consequently, the adhesins can slip out of the outer-

[1]CSSB Centre for Structural Systems Biology, Deutsches Elektronen-Synchrotron DESY, Hamburg, Germany. [2]Department of Microbiology and Immunology, Geisel School of Medicine at Dartmouth, Hanover, NH, USA. [3]Christian-Albrechts-University, Kiel, Germany. ✉e-mail: holger.sondermann@cssb-hamburg.de

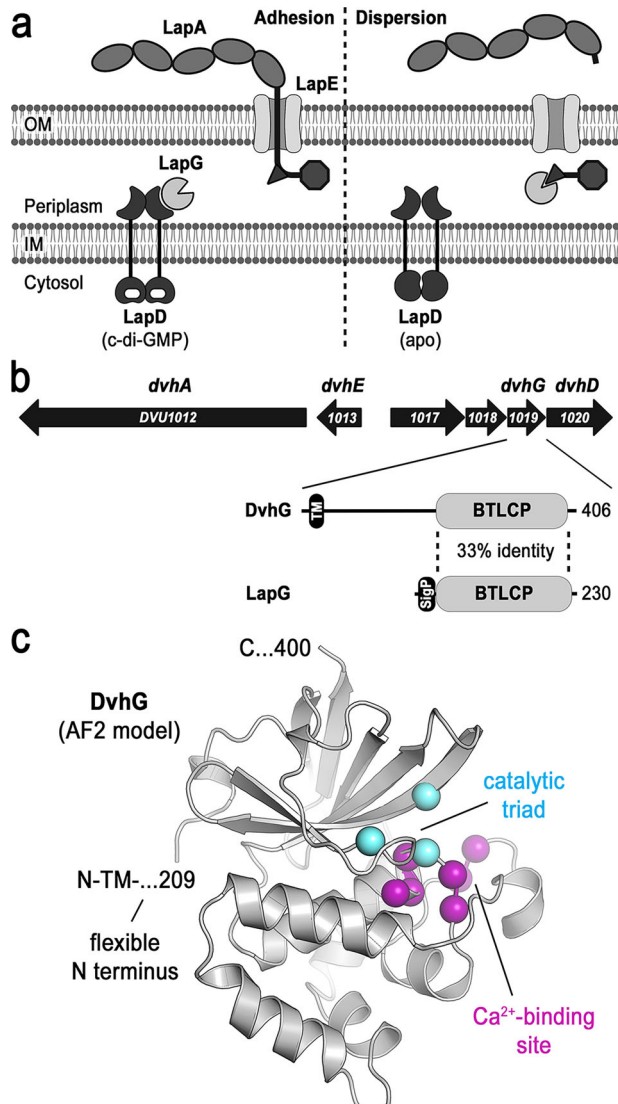

**Fig. 1 | Large adhesin regulation in Gammaproteobacteria and *D. vulgaris* Hildenborough. a** Overview of c-di-GMP- and protease-regulated cell adhesion in Gammaproteobacteria. **b** Operon structure encoding a large adhesin system in *D. vulgaris* Hildenborough. DVU1012 encodes the adhesion DvhA, DVU1013-1018 encode the corresponding T1SS. DVU1019 encodes a predicted BTLCP protease, DvhG. The identity of DVH1020/DvhD is described in detail at a later stage. **c** AF2-based model of DvhG highlighting a conserved catalytic triad and calcium-binding site. Only the conserved catalytic core of the AF2-based model is shown. Cartoon schematic in (**a**): Created in BioRender. Sondermann, H. (2026) https://BioRender.com/2hdwdgf.

membrane component of their secretion system (Fig. 1a, right), enabling cell detachment[13].

Several organisms encode two or more adhesins regulated by the same Lap signaling system[2,20]. One adhesin is usually encoded within an operon containing LapD and LapG, while others can be located elsewhere in the genome. In *P. fluorescens*, LapA is a *lap* operon-encoded adhesin, whereas adhesin MapA is associated with a dedicated secretion system in its own operon[20]. The two adhesins are found in different biofilm regions, with MapA predominantly in nutrient- and oxygen-deprived areas. Both adhesins are subject to transcriptional as well as post-translational control via c-di-GMP, LapD, and LapG[21]. In *Vibrio cholerae*, the cholera-causing agent, two adhesins controlled by LapD/LapG, FrhA, and CraA, also appear non-redundant, facilitating adhesion and biofilm formation on distinct surfaces[2,9]. In contrast to

the detailed understanding of adhesin control in Gammaproteobacteria, much less is known about these processes in organisms outside of this class.

Sulfate-reducing bacteria (SRB), such as the Deltaproteobacterium *Desulfovibrio vulgaris* Hildenborough (also known as *Nitratidesulfovibrio vulgaris* Hildenborough), thrive in the environment because of their ability to form biofilms on surfaces[22,23]. The *D. vulgaris* Hildenborough genome encodes a large c-di-GMP signaling network[24], indicating a crucial role of the second messenger and its downstream processes in the organism's physiology. The organism also expresses two adhesins required for biofilm formation: DvhA (DVU1012 or BfsA) and DVU1545 (or BfsF)[25–29]. DvhA is encoded in an operon (Fig. 1b) that resembles the Lap system in Gammaproteobacteria, including a gene for a TolC-like T1SS component, analogous to LapE, that is required for biofilm formation[25,29,30]. The operon also encodes two other proteins, DvhG (DVU1019 or BfsG) and DvhD (DVU1020 or BfsD), that have been recently described to function analogously to LapG and LapD in a reconstituted system[25,30]. In this recent study, a *P. fluorescens* strain lacking LapD and LapG was engineered to express DvhD and DvhG. In addition, the LapA segment containing the LapG-cleavage site was replaced for the analogous sequence of DvhA[30]. This reconstituted system enabled the investigation of DvhD and DvhG regulation in a *P. fluorescens* biofilm model, revealing that (i) DvhG cleaves DvhA, releasing it from the cell surface, and (ii) DvhG is regulated by DvhD in a c-di-GMP-dependent manner in cells. While these results support that DvhG and DvhD perform functions that parallel those of the *P. fluorescens*-native Lap system, it is interesting to note that DvhD and DvhG have a sequence and domain organization distinct from those of their functional counterparts in the Lap system. Specifically, DvhD contains a hitherto undefined periplasmic domain as well as cytoplasmic Per-Arnt-Sim (PAS) and HD-GYP domains[31,32]. PAS domains occur in all kingdoms of life and often possess sensory function, whereas HD-GYP domains, named after their conserved sequence motifs, bind and, in many cases, cleave cyclic dinucleotides. The DvhD domain organization contrasts with LapD's LapD/MoxY periplasmic domain, followed by a cytoplasmic, catalytically inactive HAMP-GGDEF-EAL module. Notably, GGDEF and EAL domains often function as diguanylate cyclases and c-di-GMP-specific phosphodiesterases, respectively, but in some cases also provide sensory functions, such as in the case of LapD[33]. How DvhD senses c-di-GMP levels and transduces this signal to regulate adhesion and biofilm formation in *D. vulgaris* remains poorly understood. Additionally, DvhG contains a transmembrane domain that predicts its localization at the inner membrane.

Here, we examine the differences between the Lap and Dvh systems using structural and biochemical approaches. Our results uncover the molecular underpinnings of DvhG and DvhD functions, revealing both conserved and convergently evolved regulatory principles controlling cell adhesion and biofilm formation.

## Results

### Conservation and distinct features of *D. vulgaris* DvhG

The putative role of DvhG in regulating DvhA has been discussed previously and is supported by reconstitution studies in a biofilm model system[25,30]. Within the catalytic core, DvhG and LapG share 33% sequence identity and structural modeling using Alphafold2 (AF2) revealed a conserved bacterial transglutaminase-like cysteine protease (BTLCP) fold (Fig. 1c, Supplementary Fig. 1a, and Supplementary Table 1)[34,35]. Two main features that LapG activity depends on, the catalytic triad (C-H-D) and an active site-adjacent calcium-binding motif, are strictly conserved in DvhG (Fig. 1c and Supplementary Fig. 1a, b). Based on this comparison, we predicted that DvhG is an active cysteine protease in SRB.

However, there are also striking differences between LapG orthologs and DvhG. While LapG is secreted into the periplasmic space as a soluble protein by means of an N-terminal signal peptide, DvhG is

predicted to have an N-terminal transmembrane helix followed by approximately 180 residues that, in the AF2 model, form a disordered linker segment (Fig. 1b, c). To confirm the prediction that DvhG is a membrane-anchored protein, we conducted comparative fractionation assays using LapG and LapD from *P. fluorescens* as controls for soluble and membrane proteins, respectively (Supplementary Fig. 1c). Here, we also included DvhD, the protein encoded next to DvhG in *D. vulgaris*, which was recently shown to be a regulator of DvhG, akin to LapD in the canonical Lap system[30]. The four proteins were expressed with a C-terminal monomeric super-folder green fluorescent protein (msfGFP) in *Escherichia coli* as the host, followed by fractionation into soluble and membrane fractions, gel-electrophoretic separation, and detection by in-gel fluorescence. Based on this analysis, both LapD and DvhD were found exclusively in the membrane fraction, consistent with the prediction of transmembrane helices present in these proteins. LapG resides exclusively in the soluble fraction, agreeing with its periplasmic localization. In contrast, the majority of full-length DvhG was detected in the membrane fraction, whereas fluorescent (N-terminally truncated) fragments appeared in the soluble fraction, likely due to unspecific proteolysis within the N-terminus of DvhG, a region predicted to be disordered.

In addition, gammaproteobacterial LapG orthologs contain a conserved binding pocket that accommodates a conserved tryptophan residue in the periplasmic domain of LapD, especially when the receptor is bound to c-di-GMP (Supplementary Fig. 1a, b)[15]. This pocket appears to be occluded in DvhG, correlating with the lack of sequence similarity between the periplasmic domains of LapD and DvhD[30]. Together, sequence conservation and structural modeling predict that DvhG is a bona fide ortholog of LapG-like proteases with features that distinguish DvhG from LapG orthologs.

### DvhG cleaves adhesin sequences at conserved sites

LapG proteases studied to date process their adhesin substrates in a sequence-specific manner, cleaving between the adhesin's retention module and the membrane conduit-spanning segment[2,19]. The optimal cleavage sequence for *P. fluorescens* LapG comprises a predicted (targeting) helix with low sequence conservation, followed by a conserved "DP" motif and the scissile bond between two alanine residues in the consensus "[T/A/P]AAG" motif[19]. To explore whether *D. vulgaris'* two large adhesins, DvhA and DVU1545, could be DvhG substrates[25,29], we first generated AF2 models for fragments spanning the entire length of the two adhesins, providing a view of their domain composition (Fig. 2a and Supplementary Table 1). Extracellularly, they share the presence of a serralysin-like metalloprotease domain at their C termini and more than a dozen β-strand-rich domain repeats. For DvhA, a von Willebrand factor (vWF) type A domain is inserted between repeats 14 and 15. At the periplasmic site, both adhesins are predicted to contain an N-terminal retention domain composed of β-strands followed by a predicted helix, consensus cleavage site, and a segment with no apparent secondary structure. Such a structure is reminiscent of the membrane-anchoring domain that was first described in a large adhesin from *Marinomonas primoryensis*, and that is a more broadly conserved feature of adhesins that are displayed at the outer membrane[7,8].

To test the susceptibility of the two *D. vulgaris* adhesins to be cleaved by DvhG, we used a reporter system where candidate sequences from DvhA or DVU1545, as predicted by structural and sequence information, were inserted between His₆-tagged SUMO and msfGFP moieties (Fig. 2b)[19]. Purified reporter constructs were incubated with *E. coli* lysates containing either *P. fluorescens* LapG or *D. vulgaris* DvhG and the products from this reaction were analyzed using SDS-PAGE and in-gel fluorescence. Lysates expressing either DvhG or LapG processed the putative substrate peptide of DvhA in a calcium-dependent manner, as supported by the lack of cleavage in the presence of the calcium-chelating agent EGTA. LapG processed the DvhA

peptide less efficiently than did DvhG under otherwise identical conditions (Fig. 2b). A substrate analog with a di-arginine instead of a di-alanine sequence in the DvhA "[T/A/P]AAG" motif showed a pronounced resistance to proteolysis in the presence of either protease, confirming the di-alanine sequence as the scissile bond. Similar results were obtained with reporters containing candidate sequences from the second *D. vulgaris* adhesin, DVU1545. In this case, DvhG and LapG processed the DVU1545 reporter to completion under otherwise identical assay conditions. DvhG was also able to efficiently cleave a peptide sequence derived from *P. fluorescens* LapA in a calcium-dependent manner. DvhG activity required direct calcium-binding and an intact catalytic triad, as protein variants with site-directed mutations in the predicted calcium-binding site (DvhG-D³¹⁶A/E³¹⁸A) or the central cysteine residue in the C-H-D motif (DvhG-C³¹⁷A) rendered the protease inactive (Supplementary Fig. 2).

Together, our findings show that DvhG's catalytic mechanism and substrate specificity is conserved with respect to the LapG-family of BTLCP-type proteases, even across different bacterial classes and despite only moderate sequence conservation.

### Adhesin retention domains interact with DvhG for increased proteolytic efficiency

Although mutational studies have revealed the consensus sequence in gammaproteobacterial adhesins required for maximal cleavage by LapG orthologs[17,19], the lack of structural details about the protease-substrate interactions limited our understanding of the substrate specificity-defining molecular mechanisms. To obtain structural insight into the interactions of LapG proteases with their substrates, we first modelled well-validated LapG-substrate pairs, including *P. fluorescens* LapG bound to LapA and MapA, as well as *V. cholerae* LapG bound to CraA and FrhA (Fig. 3a, Supplementary Fig. 3, and Supplementary Table 1)[2,20]. Invariably, the models showed the conserved "[T/A/P]AAG" motif, which contains the scissile bond, aligned with the catalytic triad (C-H-D) of the corresponding LapG ortholog. The cryptic helix that precedes the consensus cleavage site is docked at the β-sheet of LapG, while the conserved "DP" motif is part of a turn between the helix and "[T/A/P]AAG" motif of the protease substrate. We refer to the helix as "targeting helix," as it provides an additional docking mechanism that explains the higher efficiency of LapG on substrates that contain this segment[19]. The conformation and LapG interaction of the substrate, which spans from the targeting helix to the "[T/A/P]AAG" motif, appear highly similar in all protease-substrate pairs and virtually identical between the individual models for a particular pair, supporting a high model confidence for this region. Beyond the scissile bond, the substrates appeared to lack interactions with the protease, consistent with variable conformations and lower model confidence for this region. Overall, the structural predictions explained the substrate specificity of gammaproteobacterial LapG orthologs at the molecular level and validate such a modeling approach to assess how DvhG may interact with the adhesins in *D. vulgaris* Hildenborough.

Initial attempts to model DvhG in complex with substrate peptides comprising the targeting helix and consensus cleavage site yielded models with only moderate confidence and low reproducibility. Upon further inspection of the protein sequences, we noticed subtle but localized differences between SRB and gammaproteobacterial adhesins within their periplasmic domains (Fig. 3a). Specifically, the sequences between the targeting helices in DvhA and DVU1545 to the corresponding retention domains on one side and scissile bonds on the other were predicted to be shorter than those in *P. fluorescens* and *V. cholerae* adhesins. AF2 models of the periplasmic fragments of the SRB adhesins further suggested close proximity of the predicted targeting helices to the retention domains of DvhA and DVU1545 (Fig. 2a). Based on this analysis and to explore alternative modes of interactions, we modeled the entire N-terminal fragment of DvhA and DVU1545,

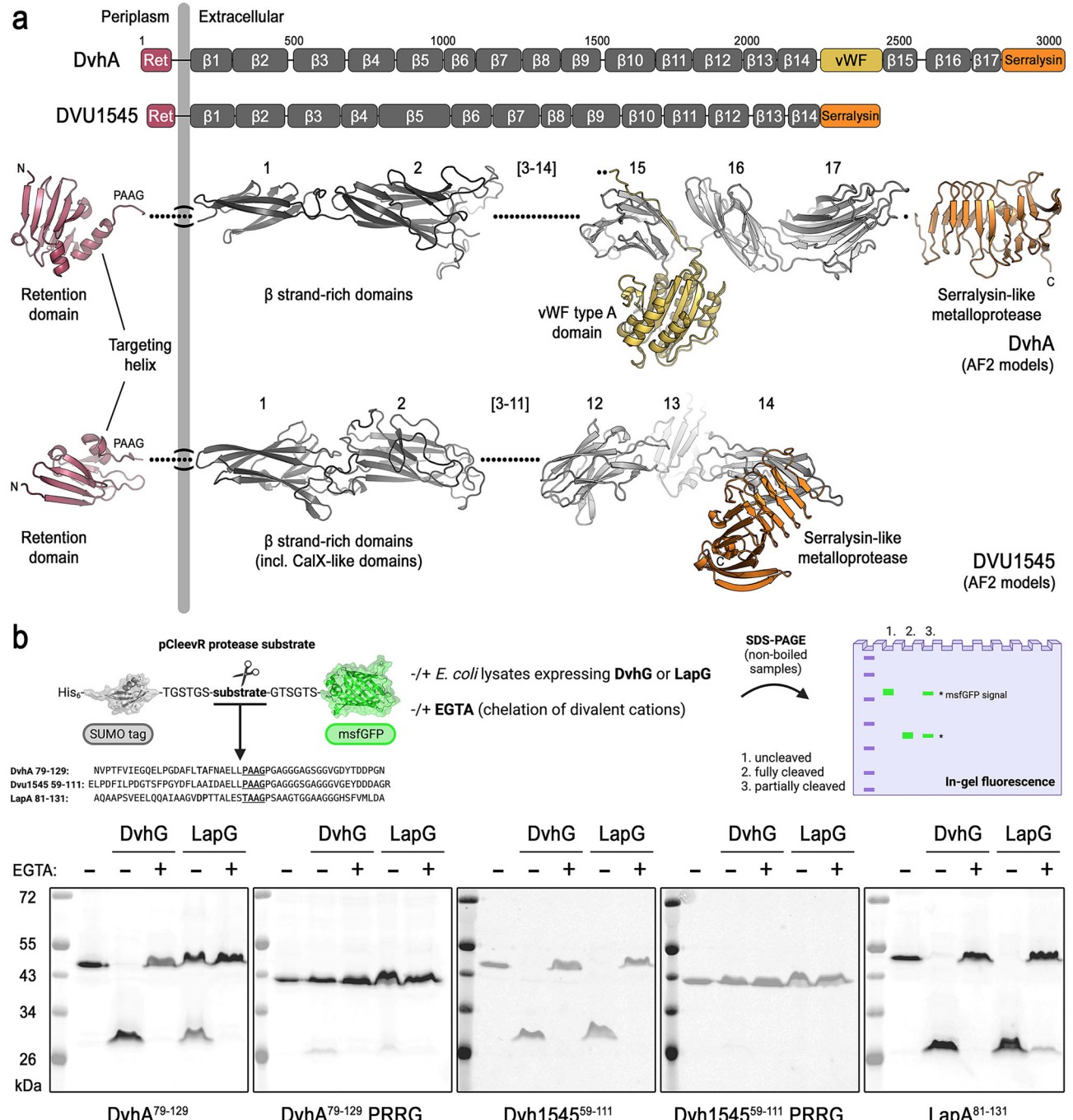

**Fig. 2 | DvhA and DVU1545 cleavage by DvhG. a** Domain organization and composite structural models for DvhA and DVU1545. The cartoons show the predicted topology of the adhesins anchored in the outer membrane. Domain boundaries were assigned based on structural models (β strand-rich domains is used as a generic term for domains predominantly composed of β strands). **b** Cleavage of consensus adhesin sequences following the retention modules by protease orthologs was assessed by a SUMO-msfGFP-based reporter system. All substrates were purified, while expressed proteases were added as cleared *E. coli* lysates containing detergent. Substrates were present at equal concentration in all reactions. Cleavage was assessed by analysis of the reaction products through in-gel fluorescence. Representative gels from two independent assays are shown. Cartoon schematic in (**b**): Created in BioRender. Sondermann, H. (2026) https://BioRender.com/k01i237.

from the retention domain to the consensus cleavage site, in complex with DvhG (Fig. 3a and Supplementary Fig. 3c). Reproducible models with high interaction scores were obtained for both pairs (ipTM scores > 0.8; Supplementary Fig. 3c and Supplementary Table 1). The scissile bond invariably lined up with the active site of the protease (Fig. 3a, b and Supplementary Fig. 3c). The targeting helix of the adhesins docked onto the bottom of DvhG's β-sheet, which involves conserved hydrophobic residues that precede the "[T/A/P]AAG", F95

and F99 of DvhA or F76 and I80 of DVU1545. Of note, the models without exception suggested an additional interface between the adhesins' retention domain and DvhG (Fig. 3a, b and Supplementary Fig. 3c). In particular, surface-exposed, hydrophobic residues on two β-strands of DvhG interact with a hydrophobic patch on the retention domain (Fig. 3b). These extensive protein-protein contacts may indicate differences in the extent to which the retention domains are involved in targeting the adhesins to specific proteases.

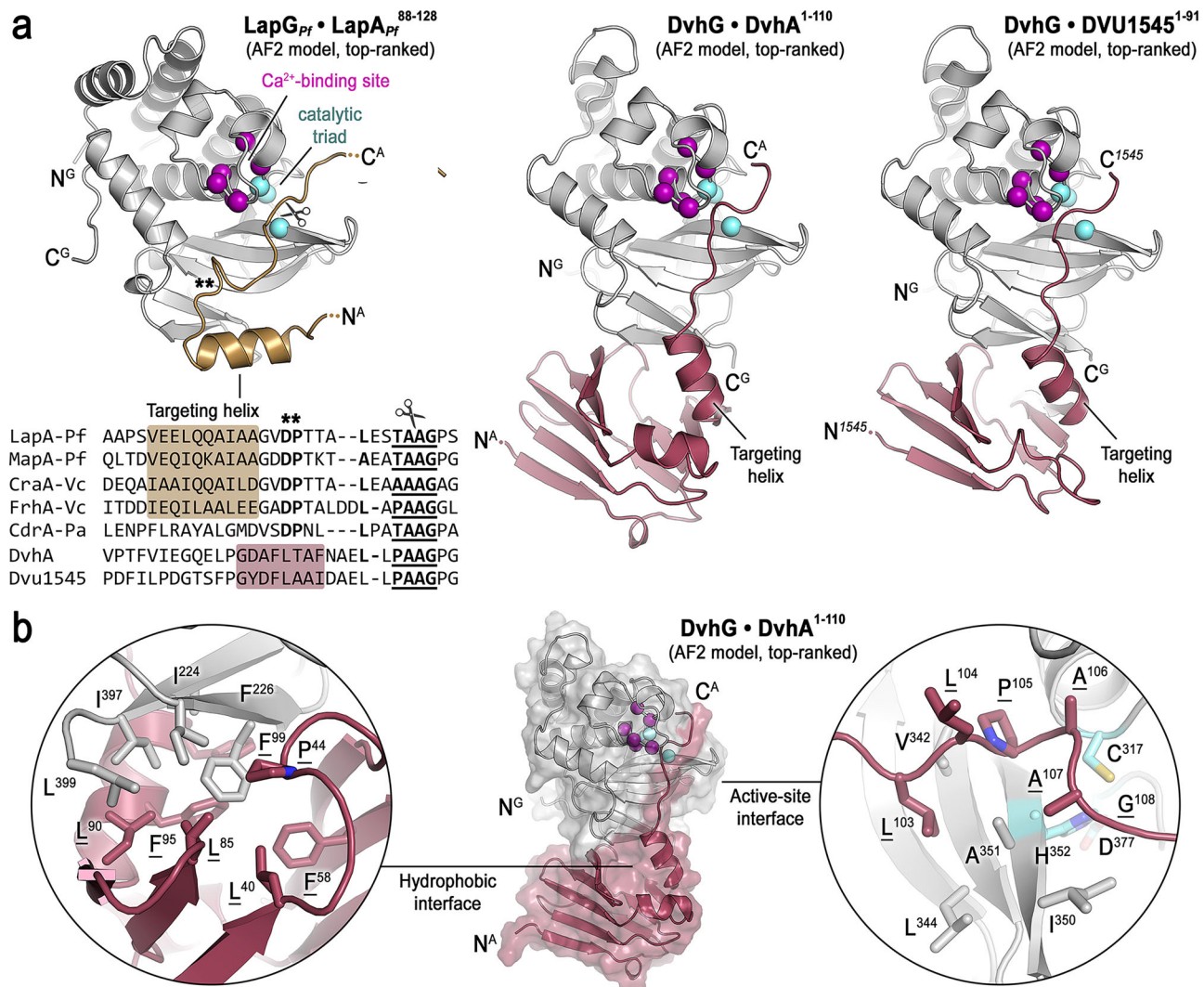

**Fig. 3 | Structural modeling of LapG and DvhG bound to substrates.**
**a** Comparison of AF2-based models for LapG$_{Pf}$•LapA$_{Pf}$, DvhG•DvhA, and DvhG•DVU1545 are depicted with the protease shown in the same orientation. The LapA sequence comprises the targeting helix and consensus cleavage site. For DvhA and DVU1545, the entire N-terminal segment up to the cleavage site was included in the modeling. Proteases are colored grey throughout. **b** A representative DvhG•DvhA complex model is shown with a transparent surface, with detailed views shown for the hydrophobic interface between DvhG and the retention modules and for the DvhG's catalytic site engaged with DvhA's scissile sequence. Adhesin residues are underlined.

To validate the structural model that the retention modules play a direct role in adhesin-protease interactions, we first analyzed the efficiency of DvhG cleavage on SUMO-msfGFP substrate reporters containing either the sequence spanning from the targeting helix to the cleavage site or the same sequence preceded by the retention domain (Fig. 4a, b and Supplementary Table 2). For both DvhA- and DVU1545-based reporters, the inclusion of the retention domain yielded enhanced processing by DvhG under otherwise identical conditions. We then mutated hydrophobic residues in DvhA on the predicted DvhG-interacting surface to charged residues (Fig. 3b). These mutant reporters purified identical to the wild-type version, but their cleavage by DvhG was affected, with mutations on the retention domain surface (L$^{90}$E or I$^{85}$E) having a bigger negative impact than a mutation in the targeting helix (F$^{95}$E) (Fig. 4c, Supplementary Fig. 4, and Supplementary Table 3).

In summary, while the basic mechanism of catalysis appears to be conserved in the BTLCP enzymes studied to date, structural modeling followed by experimental validation indicates differences in the ways substrates engage with their corresponding proteases and how substrate specificity is established.

## DvhD contains a periplasmic dCache domain

A recent study has shown that DvhD enables c-di-GMP-dependent control over DvhG in a *P. fluorescens* strain lacking LapD and LapG, supporting that DvhD could be the functional homolog of LapD in SRB[30]. However, DvhD bears no sequence or domain resemblance to LapD (Fig. 5a). Unlike LapD, which has a cytosolic S-helix/GGDEF/EAL domain module responsible for c-di-GMP binding, DvhD contains an HD-GYP domain with the potential to bind or cleave c-di-GMP[16,31]. In LapD, c-di-GMP binding to the cytosolic domain generates a conformational change that is transduced to its periplasmic LapD/MoxY domain[2,16,36]. To gain structural insight into the mechanisms that enable DvhD to sense and transduce c-di-GMP levels, we first examined an AF2-based model of DvhD (Fig. 5b, left panel and Supplementary Table 1). We opted to model DvhD as dimers, based on reports that HD-GYP domains form dimeric active assemblies[37,38]. The model predicted loosely dimeric periplasmic domains. The transmembrane domains cluster into a bundle that leads to a cytoplasmic module comprising tightly dimerized juxtamembrane helices and PAS/HD-GYP domains. A helical stalk connecting PAS with the HD-GYP domain appears as a prominent dimerization motif. Considering

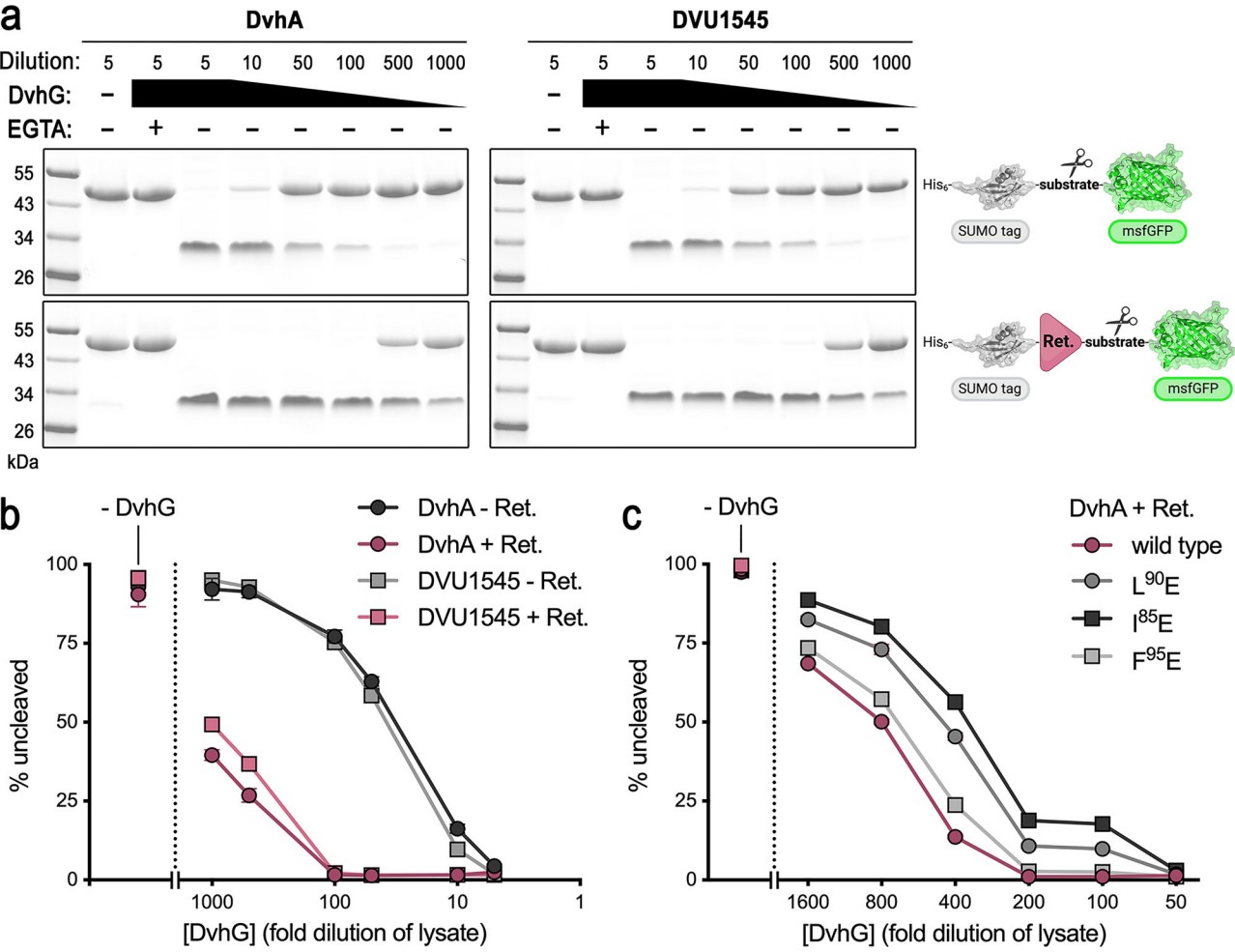

**Fig. 4 | DvhA and DVU1545 proteolysis by DvhG. a** Cleavage reactions for DvhA and DVU1545 substrate reporters containing the residues spanning the respective adhesin targeting helix and cleavage site without (top) or with (bottom) the retention domain. Substrate reporters were purified and added to serially diluted lysates from *E. coli* cultures that expressed full-length DvhG. A parallel reaction at the highest lysate concentration was supplemented with EGTA. Reactions containing *E. coli* lysates harboring an empty plasmid were added at the highest concentration only. Note that samples are not boiled to allow for msfGFP to remain folded. As a result, proteins migrate as partially folded entities, rendering size estimations based on migratory behavior unreliable. Representative gel images from four independent experiments are shown. **b** Quantitative analysis of substrate proteolysis by DvhG based on three independent experiments. Mean values and standard deviations (SD) are shown. **c** Quantitative analysis of the effect of DvhA mutations at the DvhA-DvhG interface for substrate processing by DvhG based on three independent experiments. Mean values and standard deviations (SD) are shown. Representative gel images are shown in Supplementary Fig. 4. Cartoon schematic in (**a**): Created in BioRender. Sondermann, H. (2026) https://BioRender.com/k01i237.

the moderate confidence scores of the DvhD model (Supplementary Table 1), we sought after experimental validation of its main features. We designed constructs to express and purify DvhD soluble fragments, including the isolated periplasmic domain, HD-GYP domain, and PAS/HD-GYP tandem fragment. These purified fragments were used in crystallization campaigns, yielding experimental structures in distinct states (Fig. 5b, right panel, Fig. 6, and Supplementary Table 4). The structures of the periplasmic domain and PAS/HD-GYP fragment of DvhD were determined at relatively low resolution and have R values at the upper end of the spectrum for structures determined at comparable resolutions. As a consequence, we focus our analysis on global features that can be deduced unequivocally from these experimental structures.

The crystal structure of the monomeric periplasmic domain, determined at 3.0 Å resolution using the phases obtained from single-wavelength anomalous diffraction (SAD) data, confirmed the structural prediction based on AF2 (Fig. 5c, d). Crystal structure and AF2 model superimpose with a rmsd of 0.74 Å over 1453 atoms, with the crystal structure containing two internal, disordered loop regions at

the membrane-distal tip of the domain (Fig. 5d). Comparative searches against the Protein Data Bank (PDB) using Foldseek revealed structural resemblance of the DvhD periplasmic domain to bacterial dCache domains (Supplementary Fig. 5), a similarity that was not apparent at the sequence level[39,40]. The bilobed dCache domains are part of the Cache family comprising predominantly extracytosolic ligand binding domains found in bacteria, archaea, and eukaryotes[39]. Based on a global structural alignment, the overall conformation of DvhD's periplasmic domain is most similar to those of KinD and SpdE (Supplementary Fig. 5a, b). The latter two have a long helix α1. In contrast, in DvhD, this helix is broken into two segments, with the second segment (α1') bending away from α1 (Fig. 5d), as has been observed in McpH[41–43]. CACHE domains, including the dCache subclass, function as ligand-sensing units in a multitude of signaling proteins (Supplementary Fig. 5c)[44,45]. However, the lack of discernible, unmodeled density at the putative ligand binding site of DvhD's dCache domain and the overall low sequence similarity to other dCache domains prevent us from determining whether DvhD binds an extracytoplasmic small-molecule ligand.

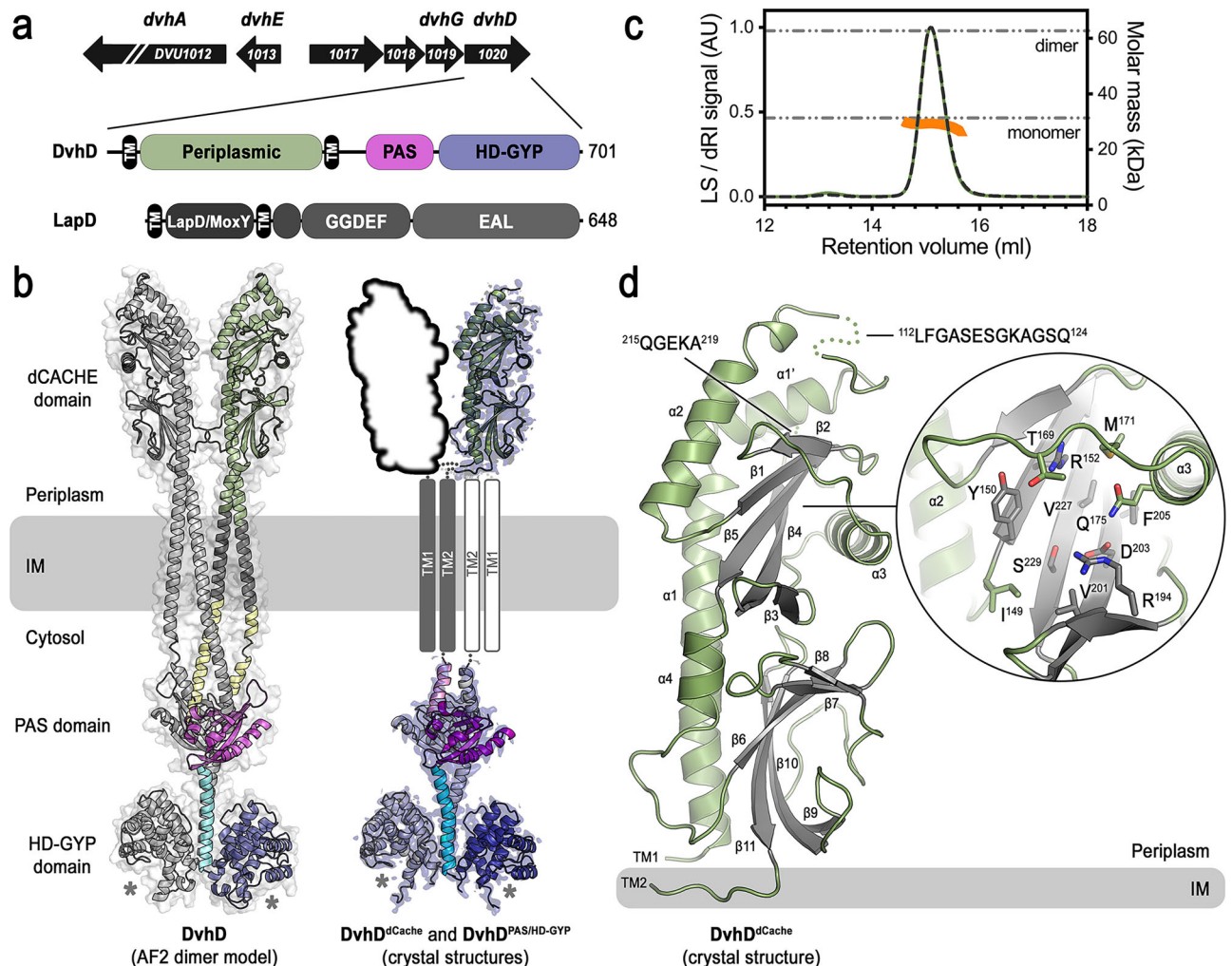

**Fig. 5 | Structural model of a DvhD dimer. a** Domain organization of DvhD and LapD. **b** AF2-based dimer model of DvhD (left panel) versus crystal structures of the isolated domains (right panel). One protomer is shown in color, the other one in grey. The AF2 model also shows the transparent accessible surface of the dimer. In the right panel, crystal structures are shown with their experimental maps (DvhD^dCache: experimental map after SAD phasing; DvhD^PAS/HD-GYP: 2Fo-Fc map). **c** SEC-MALS data of the periplasmic domain of DvhD. Dashed and solid curves show the light-scattering and refractive index signal, respectively, from a size exclusion-coupled multi-angle light scattering experiment. The data points across the peak report molar mass calculations based on light scattering. Theoretical molecular masses of a monomer and dimer based on protein sequence are indicated as horizontal lines. **d** Crystal structure of the purified periplasmic domain of DvhD. Disordered loops are indicated. The magnified cutout depicts the region of the protein, including side chains shown as sticks, which corresponds to ligand-binding sites in canonical dCache domains.

## The HD-GYP domain of DvhD serves as a c-di-GMP-binding switch module

We also determined the crystal structure of the entire cytoplasmic fragment of DvhD, comprising PAS and HD-GYP domains (Fig. 5b, right panel). The structure was solved to 3.4 Å resolution by molecular replacement with isolated domains from the full-length AF2 model as search entries (Supplementary Table 4). Given the low resolution of the electron density map and moderate refinement statistics, we focus here mainly on the global structural arrangement. Overall, the AF2 model of the full-length DvhD dimer depicted the arrangement of the domains and linker segments in the crystal (Fig. 5b). However, the final crystallographic model differed from the AF2 model in the exact positioning of the two PAS and HD-GYP domains within the dimer and the relative position of the stalk linking the two domains within a monomer. Dimerization in the crystallographic model is driven mainly by the stalk helices linking the PAS and HD-GYP domains, without prominent intra- or intermolecular interactions outside the stalk-like helical elements. The putative c-di-GMP-binding sites in the HD-GYP domains face outwards and are solvent-accessible (Fig. 5b, asterisks).

However, it remained unclear whether DvhD binds to and potentially hydrolyzes cyclic dinucleotides.

To characterize nucleotide binding to DvhD and enable comparisons with other HD-GYP domain-containing proteins, we first determined the crystal structure of the isolated HD-GYP domain in the presence of c-di-GMP (Fig. 6 and Supplementary Table 4). The structure was solved to 1.8 Å by molecular replacement using the AF2-derived HD-GYP domain model as the input, resulting in placement of four molecules in the asymmetric unit of the crystal, arranged into two dimeric assemblies. The position of the stalk helices that would lead to the PAS domains required manual adjustment. Upon refinement, density at the putative nucleotide binding sites was clearly defined and could be modeled as c-di-GMP molecules (Fig. 6a, right panel). Surprisingly, a density that fits a third molecule of c-di-GMP per dimer was observed at the bottom of the helical stalks, forming a bridge within a HD-GYP domain dimer (Fig. 6b). We refer to these sites as "canonical" and "bridging" c-di-GMP-binding sites, respectively (Fig. 6a).

At the canonical site, residues $Y^{652}$ and $L^{577}$ clamp one of the aromatic nucleobases of c-di-GMP, whereas $M^{646}$ buttresses the other

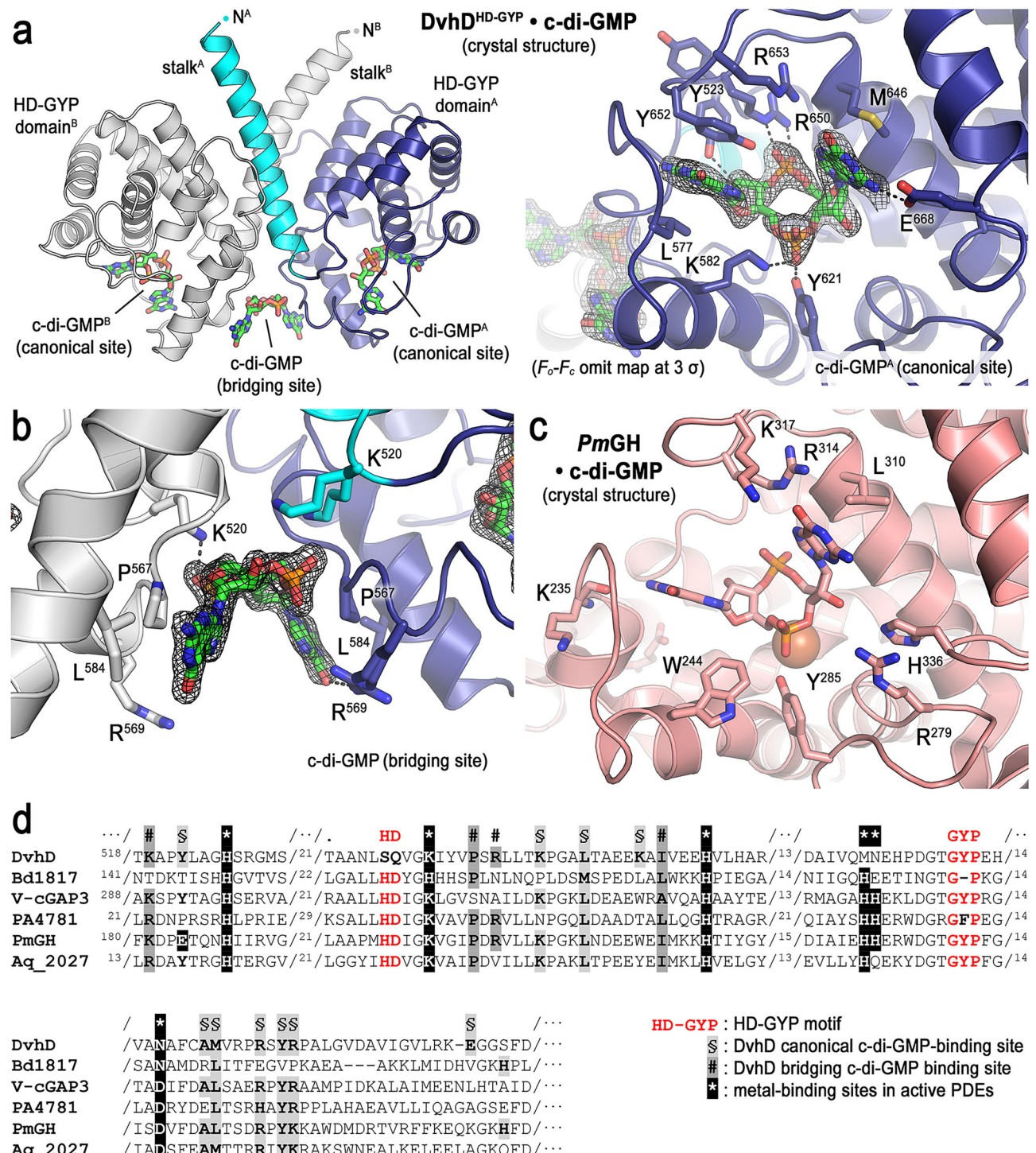

**Fig. 6 | Structure of the c-di-GMP-bound HD-GYP domain of DvhD. a** Overview of a crystallographic HD-GYP domain dimer bound to three molecules of c-di-GMP. One HD-GYP monomer is shown in white, the other one in color. The canonical binding site is shown in a detailed view with residues that interact with c-di-GMP shown in sticks and with an omit map density for c-di-GMP, contoured at 3 σ (right panel). **b** A detailed view of the bridging site is shown with an omit map density for c-di-GMP, contoured at 3 σ. **c** The HD-GYP domain of *Pm*GH bound to c-di-GMP (pdb 4MDZ[37]) is shown in the same orientation as DvhD's c-di-GMP-bound HD-GYP domain in (**a**). **d** Multi-sequence alignment of HD-GYP domains with experimentally determined structures. Functionally important sites are shaded and annotated.

guanine moiety (Fig. 6a). Residue E[668] forms a hydrogen bond with the amino group of the second base, providing a mechanism for nucleotide specificity at this site. Y[621] and K[582] engage one phosphodiester bond of c-di-GMP via hydrogen bonds, whereas the second phosphodiester bond of the nucleotide forms hydrogen bonds with R[650]. The side chain of Y[523] forms a hydrogen bond with the hydroxyl group of one of the ribose groups in c-di-GMP. In contrast, at the bridging site,

specific polar side chain-nucleotide interactions were minimal and weak, involving only K[520] close to the phosphodiester bonds of c-di-GMP and R[569] at the periphery of the complex (Fig. 6b). Nucleotide binding involved symmetric proline-aromatic interactions between P[567] of DvhD and the nucleobases of c-di-GMP, with shape complementarity between the HD-GYP domain dimer and a curved signaling nucleotide being the most obvious feature (Fig. 6b and

Supplementary Fig. 6a). The structure of an unrelated GAF/HD-GYP domain-containing protein, *Pm*GH, determined in an independent study, showed a HD-GYP domain dimer similar to that observed for the PAS/HD-GYP domain fragment (Supplementary Fig. 6b)[37]. Although the structure of *Pm*GH was determined with a c-di-GMP molecule bound to one of the active sites of the protein dimer, the corresponding site occupied by the bridging c-di-GMP molecule in DvhD was occluded. Hence, it is not clear whether the latter site is an innovation specific to DvhD or whether the structure of *Pm*GH depicts a sub-stoichiometric state.

Phosphodiesterases with HD-GYP domains have been shown to require metal cations, including $Mg^{2+}$, $Mn^{2+}$, or $Fe^{2+}$, for enzymatic activity[24,31,37,38,46–49]. No density accounting for metal ions at the active site of DvhD was observed, suggesting that nucleotide-binding is independent of metal cofactors. A comparison with *Pm*GH bound to metal ions and nucleotide substrate revealed differences in the conservation and position of key metal-coordinating residues (Fig. 6c)[37]. Notably, the name-giving HD motif, which coordinates a ferrous ion in *Pm*GH, has the sequence SQ in DvhD. The double-histidine motif in *Pm*GH contributes additional side chains for iron coordination, which is non-conservatively substituted by methionine and asparagine residues in DvhD. The lack of crucial metal-coordinating residues in DvhD was also apparent in a sequence alignment comparing HD-GYP domain-containing proteins, including active phosphodiesterases, suggesting that DvhD is incapable of turning over substrate in a metal-dependent manner (Fig. 6d). This prediction was confirmed by in vitro assays, in which DvhD's HD-GYP domain or the PAS/HD-GYP tandem module failed to hydrolyze c-di-GMP regardless of the added ions (Supplementary Fig. 7). Together, these results indicate that DvhD functions as a c-di-GMP-binding protein, akin to the c-di-GMP receptor LapD.

Next, we investigated whether DvhD undergoes any conformational change upon c-di-GMP binding (Fig. 7). A side-by-side comparison of the nucleotide-free PAS/HD-GYP and the c-di-GMP-bound HD-GYP domain structures revealed two major differences between the two states (Fig. 7a): (i) The helical stalks connecting the HD-GYP catalytic domain to the PAS domain form a tight dimer in the apo conformation. The dimerization interface extends to the PAS domain and juxtamembrane helices. In contrast, in the c-di-GMP-bound conformation of the HD-GYP domain, the helical stalks are splayed apart, spanning a distance at their distal tips that is ~10 Å longer than in the apo state (Fig. 7a). This was realized through the pivoting of the helix relative to the HD-GYP domains (Fig. 7b). Although crystallization may have an influence on protein conformation, crystal packing interfaces are rather weak (calculated $\Delta^i G$ values of −0.9 to −4.8 kcal/mol for interfaces involving the stalks; compared to dimer interfaces with $\Delta^i G$ values of −21.3 and −24.7 kcal/mol)[50]. (ii) The loops of the HD-GYP domains in the DvhD dimer, which determine the shape and interactions with the bridging c-di-GMP molecule at the bottom of the helical stalks, appear partially disordered in the c-di-GMP-unbound conformation (Fig. 7a). Notably, the canonical and bridging c-di-GMP-binding sites are adjacent to each other with the loops contributing to both sites (Fig. 7b), which may suggest co-occurring or cooperative c-di-GMP binding to DvhD. According to SEC-MALS analysis, the soluble PAS/HD-GYP protein forms a constitutive dimer that is not impacted by c-di-GMP addition (Fig. 7c). However, the isolated HD-GYP domain is monomeric in the absence of c-di-GMP, but dimerizes when c-di-GMP is provided in the mobile phase of the gel filtration, indicating an impact of the second messenger on domain interactions within DvhD.

In summary, the structural analysis reveals a role of the helical segments preceding the HD-GYP domain in receptor dimerization, with c-di-GMP introducing a major conformational change that leads to altered inter-domain interactions. This change, which drives the central helical stalks apart, is likely to affect the dimerization of the PAS domain and the relative position of the juxtamembrane helices. Because these helices connect to transmembrane helices and

periplasmic domains, our structural analysis supports that c-di-GMP binding to DvhD cytoplasmic domains triggers a series of "inside-out" conformational changes that likely impact periplasmic outputs, a signaling mechanism that mimics the molecular function of LapD in Gammaproteobacteria[16,17,51].

## C-di-GMP is the preferred ligand for the HD-GYP domain of DvhD in solution

Structural analysis indicated a specific mode of c-di-GMP binding at two distinct sites in the HD-GYP domain of DvhD. Previous studies have shown that enzymatically active HD-GYP domains not only catalyze the first step of c-di-GMP degradation, that is, conversion to linear di-GMP (pGpG), but also, in some cases, the second step to two molecules of GMP[24,37,46,52,53]. Hence, we cannot rule out pGpG as a ligand for the HD-GYP-like domain of DvhD. Furthermore, some deltaproteobacterial genomes encode enzymatic entities that may produce 3'3'-cGAMP, another bacterial signaling nucleotide with distinct physiological roles[54]. Therefore, given their structural similarity, 3'3'-cGAMP could potentially bind to the same sites as c-di-GMP.

To test the relative binding of nucleotides to the purified HD-GYP and PAS/HD-GYP domains of DvhD, we quantified the effect of nucleotides on the thermal stability of proteins using nano differential scanning fluorimetry (nanoDSF) (Fig. 8 and Supplementary Fig. 6c). Both proteins were stabilized by c-di-GMP in a concentration-dependent fashion (Fig. 8a). Little or partial stabilization was observed at sub-stoichiometric c-di-GMP concentrations of 70 and 230 μM, respectively. At saturating c-di-GMP concentrations, the entire protein population shifted to increased thermal stability, exhibiting melting temperature shifts of 22 °C and 14 °C for the HD-GYP and PAS/HD-GYP fragments, respectively (Fig. 8a, b). The PAS/HD-GYP fragment experienced comparable maximal stabilization in the presence of c-di-GMP and 3'3'-cGAMP (Fig. 8b and Supplementary Fig. 6c). Linear di-GMP (pGpG) showed an intermediate level of stabilization, whereas c-di-AMP and linear AMP-GMP (pApG) were less effective under comparable conditions. The isolated HD-GYP domain showed more pronounced discrimination between c-di-GMP and 3'3'-cGAMP than the PAS/HD-GYP fragment, and stabilization by pGpG was lower for the single domain than for the tandem domains (Fig. 8b and Supplementary Fig. 6c).

Results are consistent with ITC data that establish c-di-GMP as the ligand with the highest affinity for the PAS/HD-GYP fragment (average effective $K_d = 99 \pm 7$ nM), followed by 3'3'-cGAMP (average effective $K_d = 501 \pm 9$ nM) (Supplementary Fig. 6d and Supplementary Table 5)[30]. Both ligands bind with stoichiometries of >1 (average $N = 1.24 \pm 0.24$ for c-di-GMP; average $N = 1.33 \pm 0.36$ for 3'3'-cGAMP), consistent with the structural data that indicated a 2:3 stoichiometry of a HD-GYP•c-di-GMP complex (Fig. 6a). ITC data for pGpG and c-di-AMP confirmed poor binding of these ligands to the PAS/HD-GYP fragment under comparable conditions, associated with large errors (for pGpG) or precluding a quantitative analysis altogether (for c-di-AMP). Together, these data indicate c-di-GMP as the preferred ligand for DvhD, but do not rule out a potential response via 3'3'-cGAMP.

Using nanoDSF, we also assessed the effect of c-di-GMP on the thermal stability of the purified HD-GYP domain harboring point mutations that target the canonical and/or bridging c-di-GMP-binding sites (Fig. 8c). We found that mutations in individual residues either at the canonical or the bridging site affected the ability of c-di-GMP to stabilize the protein. Specifically, mutations $R^{650}A$ (canonical site) and $P^{567}T$ (bridging site) resulted in destabilization compared to the wild-type protein when c-di-GMP was added. A similar effect was observed when simultaneously mutating HD-GYP residues $Y^{652}A/R^{653}A$ (canonical site). Other mutations had smaller effects on protein stability. Specifically, mutations $M^{646}A$ (canonical site) or $K^{520}E$ (bridging site) only slightly affected thermal stabilization by c-di-GMP. However, mutating simultaneously these two residues (HD-GYP-$M^{646}A/K^{520}E$), showed a

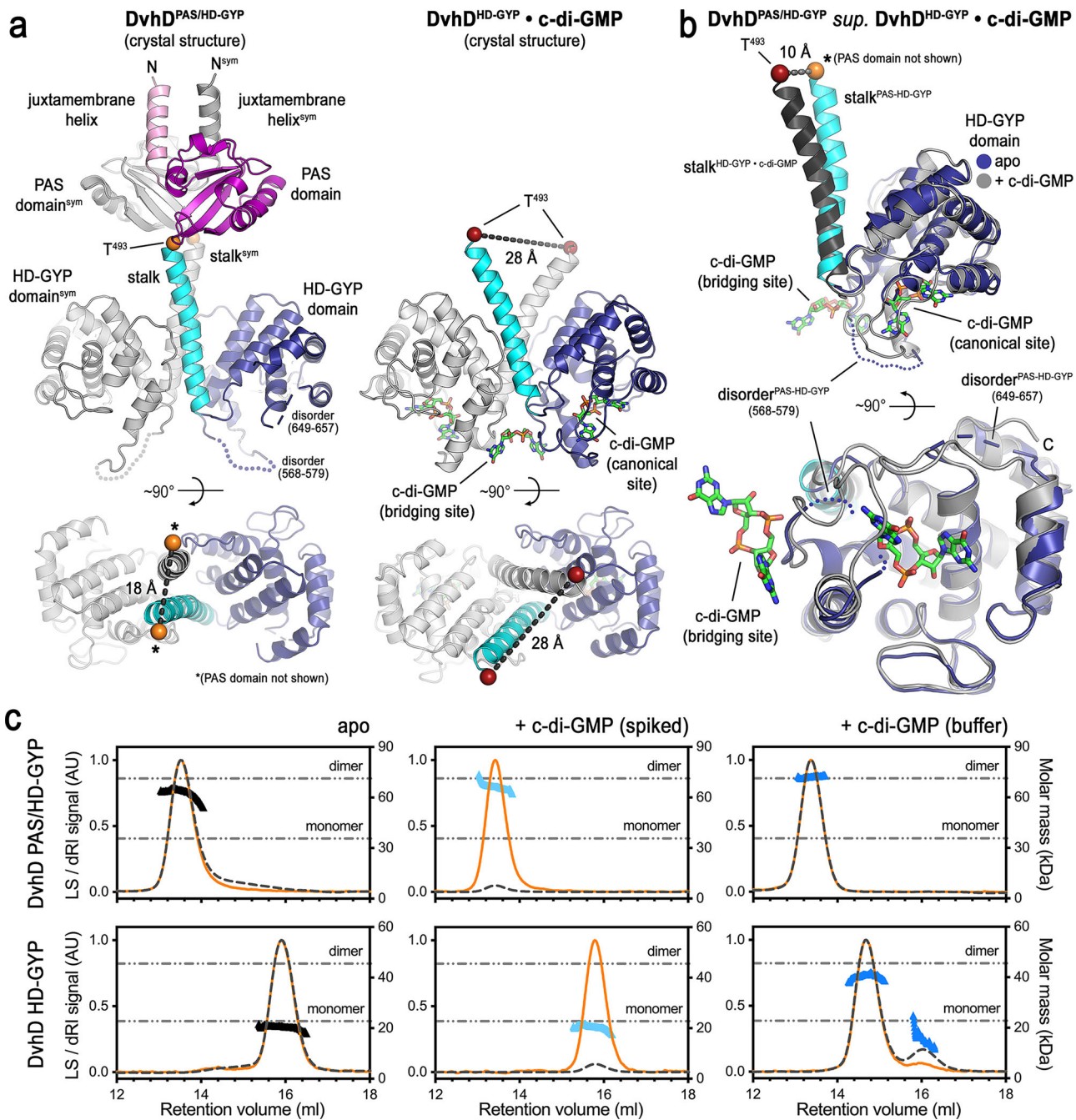

**Fig. 7 | Conformational changes in a DvhD dimer upon c-di-GMP binding. a** Side-by-side comparison of the crystallographic PAS/HD-GYP dimer (left) and c-di-GMP-bound HD-GYP dimer (right) in two orthogonal views are shown. The PAS domain was omitted for clarity in the top view of the PAS/HD-GYP dimer. Distances were measured at identical Cα positions at the distal tips of the helical stalks. **b** Stalk/HD-GYP monomers were extracted from both crystal structures shown in (**a**) and superimposed using the HD-GYP domains as reference. C-di-GMP binding sites and displacement of the helical stalks are indicated. **c** SEC-MALS data for DvhD PAS/HD-GYP (top) and HD-GYP (bottom). SEC-MALS traces and molar mass calculations are shown. Dashed and solid curves show the light-scattering and refractive index signal, respectively, from a size exclusion-coupled multi-angle light scattering experiment. The data points across the peak report molar mass calculations based on light scattering. Theoretical molecular masses of a monomer and dimer based on protein sequence are indicated as horizontal lines. Proteins were analyzed in the absence of c-di-GMP, preincubated with c-di-GMP but without nucleotide in the SEC mobile phase, or preincubated with c-di-GMP followed by SEC with a mobile phase containing c-di-GMP.

stronger effect on c-di-GMP-mediated stabilization than either single mutant alone, suggesting that c-di-GMP binding at both canonical and bridging sites is important for DvhD function. Notably, none of the point mutations examined affected the stability of the protein in the absence of c-di-GMP, but all of them negatively impacted the c-di-GMP-dependent dimerization of the HD-GYP domain (Supplementary Fig. 8). In the case of mutant P[567]T (bridging site), we also observed an

increased propensity of the HD-GYP domain to form homodimers in the absence of c-di-GMP, suggesting this region of the protein is involved in controlling inter-molecular domain interactions. Together, these results validate the structural observations that the HD-GYP domain of DvhD binds c-di-GMP through two distinct binding sites and demonstrate that c-di-GMP binding to these sites impacts the dimerization state of DvhD domains.

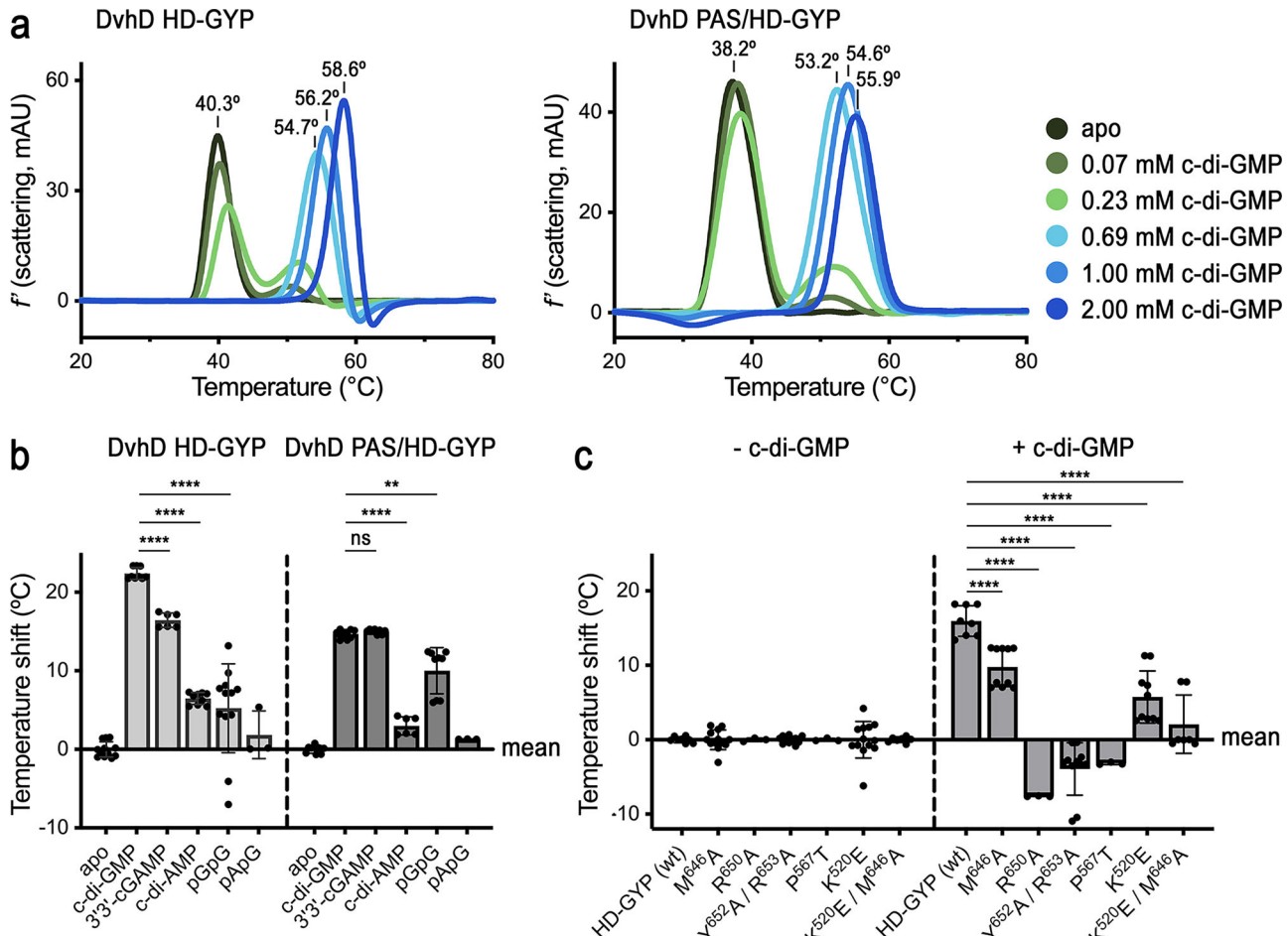

**Fig. 8 | Nucleotide binding to DvhD's cytosolic domain. a** Thermostability of the HD-GYP and PAS/HD-GYP fragments in the presence of increasing amounts of c-di-GMP is shown, measured by nanoDSF. Data are shown as the first derivative of the scattering curve upon heat denaturation. Peaks correspond to the major transitions during thermal denaturation of the proteins (main-peak temperatures are indicated). **b** Quantification of HD-GYP and PAS/HD-GYP stabilization by different nucleotides under saturating conditions (see Supplementary Fig. 6C for representative melting curves). Data are expressed as stabilization compared to the melting transition of the protein in the absence of ligand. Data are shown as mean with SD from at least three technical replicates (number of replicates are listed in the source data). The mean value calculated for the wild-type, apo protein was subtracted from the individual values derived for proteins incubated in the presence of the indicated nucleotides. Statistical significance was determined using a two-tailed Student's *t*-test (ns: $P < 0.05$; **: $P < 0.01$; ****: $P < 0.0001$). **c** Quantification of the effect of point mutation on the thermostability of the HD-GYP domain in the absence and presence of c-di-GMP under saturating conditions. The mean value calculated for the wild-type or mutant, apo protein was subtracted from the individual values derived for corresponding protein variants incubated in the presence of the indicated nucleotides. Data are shown as mean with SD from at least three technical replicates (number of replicates are listed in the source data). Statistical significance was determined using a two-tailed Student's *t*-test (****: $P < 0.0001$).

## Discussion

Biofilm formation is an integral part of the physiological response during the life cycle of many bacterial organisms, allowing them to persist in hostile environments or capitalize on beneficial niches. The Lap system, which comprises a large adhesin and a protein module dedicated to its transport and c-di-GMP-dependent regulation, has emerged as a common signaling unit for biofilm formation in *P. fluorescens* and many other species[2,31,55–57]. Here, we studied an evolutionarily distinct system in *D. vulgaris*, uncovering similarities and differences to the well-characterized Lap systems (Fig. 9a). Notably, this system is not restricted to *D. vulgaris* Hildenborough, but is also encoded in the genomes of other Desulfovibrionales, including the commensal bacteria *D. piger*, *D. desulfuricans*, and *D. fairfieldensis* (Figs. 9b and 10a, b and Supplementary Data 1)[30,58,59]. While present in healthy individuals where these organisms likely play a role in gut health due to their ability to produce hydrogen sulfide and exert pro-inflammatory effects, these bacteria have been also occasionally associated with bacteremia and inflammatory bowel disease[58,60–62]. The mechanistic details controlling commensal or pathogenetic growth of

*Desulfovibrio* strains are not well known. One may speculate that the regulated deployment of cell adhesion factors, potentially involving DvhDG-containing signaling hubs, play a major role in healthy colonization or dysbiosis.

We demonstrate that the periplasmic protease DvhG is active against specific sequences in the N-terminal domain of both *D. vulgaris* adhesins, DvhA and DVU1545[25,29]. Similar to other Lap systems, we found that the scissile bond follows the predicted periplasmic retention domains, a widespread feature in adhesins of Gram-negative bacteria. However, a distinct feature of the *D. vulgaris* system is that the interactions between DvhG and the adhesins extends further, also involving the retention domains that dock onto the protease, increasing proteolytic efficiency (Fig. 9a). Mapping sequence conservation across homologous Desulfovibrionales proteins onto the structural model of DvhG bound to DvhA revealed that conserved residues cluster at the active and calcium-binding sites but also at the site where DvhA's retention domain docks onto DvhG (Fig. 10a). Our analysis also suggests that DvhG and LapG differ with respect to their interaction with the corresponding c-di-GMP receptors, DvhD and

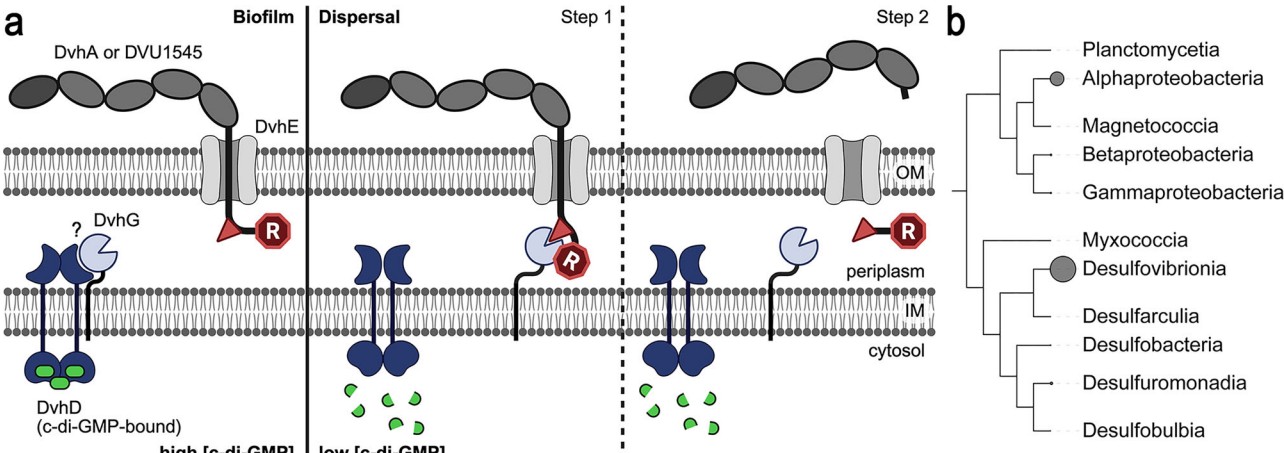

**Fig. 9 | Model and phylogenetic distribution of the DvhDG signaling system.**
**a** Overall model. At high cellular c-di-GMP levels, DvhD restrains the activity of DvhG by a hitherto unknown mechanism. As a result, adhesins are stably anchored in the outer membrane, mediating biofilm formation. When c-di-GMP levels drop in the cytosol, DvhG is released and able to engage adhesin via their retention domain and cleavage motif, resulting in a subsequent step in proteolytic release of the retention domain and liberating the extracellular adhesin fragment from the cell surface, ultimately dispersing biofilms. **b** Phylogenetic distribution of DvhDG shown at the class level. The size of the circles next to the classifications scales with the number of strains in a certain class containing a DvhDG system. Cartoon schematic in (**a**): Created in BioRender. Sondermann, H. (2026) https://BioRender.com/2hdwdgf.

LapD, respectively. In the canonical LapD-LapG system, LapG engages with high affinity with c-di-GMP-bound LapD, docking onto a specific, conserved tryptophan residue in LapD's periplasmic domain. This interaction has been proposed to sequester LapG at the inner membrane, preventing its ability to cleave adhesins[15–17,19,51]. Here, we show that DvhG contains a transmembrane domain and is membrane localized when expressed in *E. coli*, even when DvhD is absent. However, the majority of DvhG orthologs are predicted to have signal peptides, paralleling the secretion mechanism of LapG in Gammaproteobacteria (Supplementary Tables 6 and 7)[63,64]. Therefore, DvhG transmembrane domain might be a distinct feature of *D. vulgaris* Hildenborough (Supplementary Table 6)[63].

Additionally, DvhD lacks an obvious DvhG-interacting (tryptophan) residue in its periplasmic dCache domain, and a DvhG structural model suggests the absence of a hydrophobic tryptophan-binding pocket akin to that of LapG (Supplementary Fig. 1b). Together, these results suggest that DvhG might not need to interact with DvhD for membrane localization and that the regulation of DvhG by DvhD, which is apparent in the reconstituted system in *P. fluorescens*[30], involves mechanisms distinct from those described for the LapG-LapD interaction. A high-confidence model of a DvhD^dCache•DvhG^BTLCP complex generated with AlphaFold 3 implicates a conserved interface involving helix α1′ of the dCache domain and a surface at the back of DvhG's active site (Fig. 10c and Supplementary Table 1). The latter surface partially overlaps with the proposed substrate binding path, suggesting that DvhD and substrate binding may be mutually exclusive (Fig. 10d). In addition, the top-scoring structural model shows a disrupted calcium-binding site in DvhG when the protease is engaged with DvhD. These mechanistic hypotheses await experimental validation but present a feasible scenario backed by sequence conservation.

Our structural analysis shows that, despite a distinct domain organization, DvhD functions in analogous ways to LapD, acting as a c-di-GMP sensor through its HD-GYP domain. Of particular interest is the ability of the HD-GYP domain dimer to bind to three c-di-GMP molecules. Previous isothermal titration calorimetry (ITC) experiments reported a high, sub-micromolar affinity of the purified PAS/HD-GYP protein fragment for c-di-GMP[30], significantly higher than that determined for LapD[16]. Notably, the reported stoichiometry of the interaction was 1.4 ± 0.02 in solution, which, though puzzling at the time, is in agreement with stoichiometry deduced from the crystal structure reported here. Based on the structural analysis and c-di-GMP-

dependent dimerization of the HD-GYP domain, we suspect that c-di-GMP at the bridging site has a stabilizing role, locking a particular HD-GYP domain dimer conformation in place. Dimerization and overall c-di-GMP binding is sensitive to mutations at both the canonical and bridging site (Fig. 8c and Supplementary Fig. 8), suggesting a cooperative binding event that is coupled with conformational and protein-protein interface changes. Such an interpretation is also consistent with ITC data, that fit best to a single-binding event model, with cooperative c-di-GMP binding resulting in lower $K_d$ values, but potentially masking single-site thermodynamics. In LapD, we did not observe a bridging c-di-GMP binding site in the corresponding c-di-GMP binding EAL domain. Instead, c-di-GMP-induced switching of LapD involves the release of autoinhibitory interactions and the dimerization of its c-di-GMP-bound EAL domains[16,51].

While c-di-GMP binds to DvhD with high affinity and alters DvhD's functional state in cells[30], we show here that the HD-GYP domain of DvhD, in principle, can also accommodate 3′3′-cGAMP and—with much lower affinity—pGpG and c-di-AMP. To our knowledge, only c-di-GMP signaling systems have been identified in *D. vulgaris* Hildenborough so far. However, one source of 3′3′-cGAMP are hybrid cyclic dinucleotide-producing and promiscuous substrate-binding (Hypr) GGDEF enzymes that are found in other Deltaproteobacteria[54]. Whether 3′3′-cGAMP is a physiologically relevant ligand for DvhD in these organisms remains to be established.

Crystal structures suggest that DvhD's HD-GYP domain undergoes a conformational change upon binding c-di-GMP, splaying the stalk-like helices connecting the PAS and HD-GYP domains. We propose that, similar to LapD, this conformational change extends through the transmembrane helices and periplasmic domain of DvhD, ultimately controlling DvhG activity[30]. Sequence conservation across homologous proteins displayed on the surface of a DvhD dimer model revealed that conserved residues coalesce along the entire DvhD dimerization axis (Fig. 10b)[65]. This includes conserved clusters at the dimer interface around the PAS domain, helical stalk segments, and HD-GYP domains, but also functional sites involved in stabilizing a dimer of the cytoplasmic domains, especially the canonical and bridging c-di-GMP-binding sites. A conserved, solvent-exposed patch at the membrane-distal tip of the dCache may point to a functional site, potentially for DvhG recruitment (Fig. 10c), while other solvent-exposed surfaces along the back-site of the DvhD dimer show significantly lower degree of conservation (Fig. 10c).

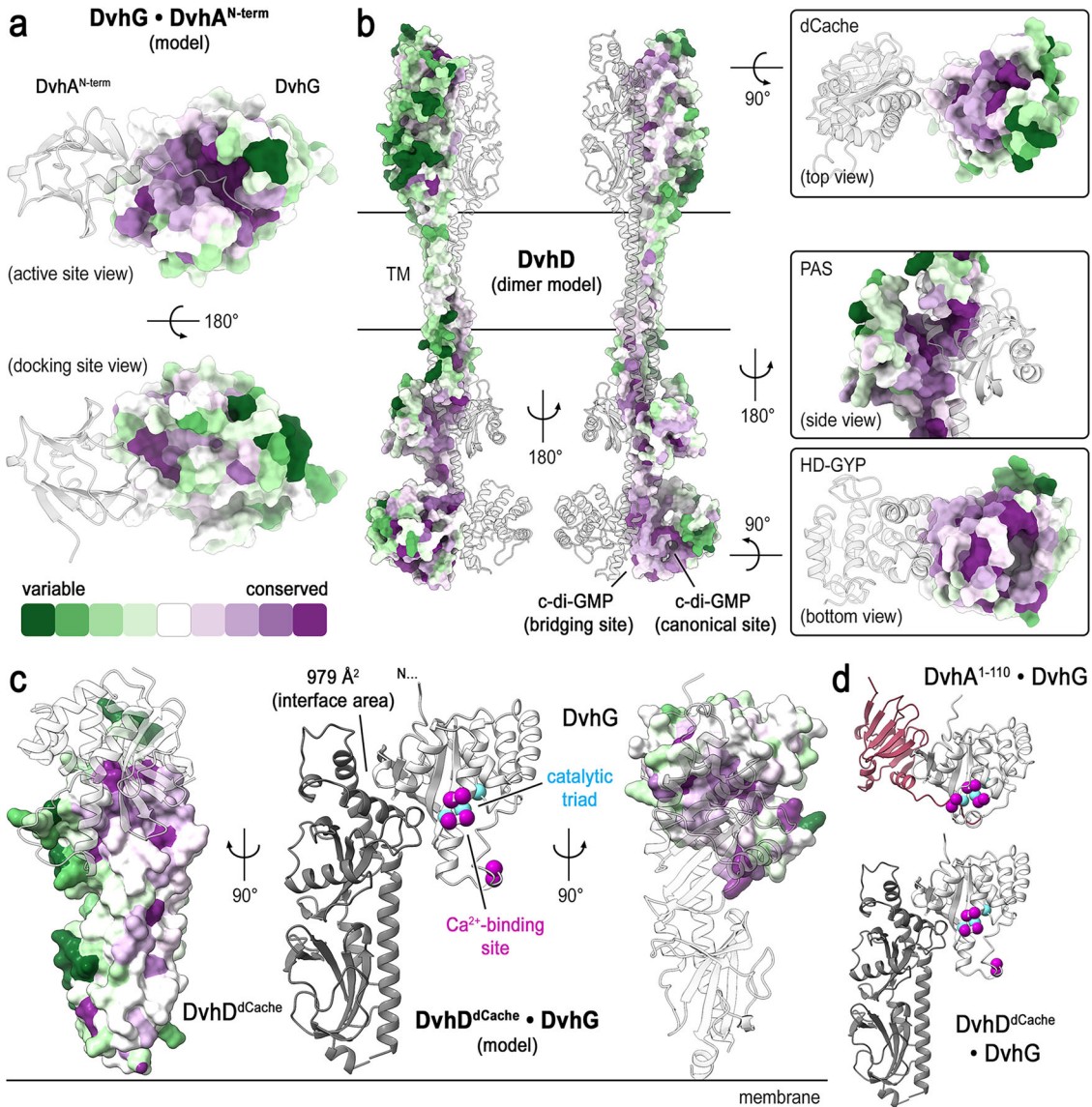

**Fig. 10 | Conservation of adhesin control in sulfate-reducing bacteria.**
**a** Conservation of Desulfovibrionales DvhG. Conservation scores were mapped onto the solvent-accessible surface of the BTLCP domain of DvhG orthologs, here modeled bound to the retention domain and cleavage sequence of DvhA.
**b** Conservation of Desulfovibrionales DvhD. Conservation scores were mapped onto the solvent-accessible surface of one half of a DvhD dimer model. Zoom-in views show conservation clusters at in the functional domains of DvhD proteins.
**c** Structural model of the DvhD dCache domain in complex with the BTLCP domain of DvhG (center). Active and calcium-binding site residues of DvhG are indicated as colored spheres at corresponding Cα atom positions. Conservation of the interface on the dCache domain (left) or the BTLCP domain (right) are shown.
**d** DvhG•substrate and DvhG•DvhD complexes are shown, with DvhG shown in the same orientation for a comparison of complex topologies.

While DvhD bears a radically different architecture to LapD, other components of the Dvh system (DvhG, DvhA, and T1SS) are similar to their counterparts in the Lap system, both in sequence conservation and function[25,29,30]. Therefore, it seems unlikely that the entire system is a case of convergent evolution. One possibility is that there was a horizontal gene transfer event from some organism that had the Lap system, but that the LapD component was not transferred or was lost. DvhD could have evolved from a HD-GYP domain-containing phosphodiesterase, either host-resident or exogenously acquired, to replace LapD. By uncovering and characterizing the similarities and differences between the Dvh and Lap systems, our structural studies contribute to a better understanding of how SRB repurposed enzymatic domains for the acquisition of distinct cell adhesion and biofilm formation regulatory mechanisms. Further study of other organisms in the *Desulfovibrio* genus, their phylogeny, and biofilm formation strategies will help elucidate the evolution of this distinct system.

Beyond this genus, other bacteria may have evolved systems that are functionally analogous to LapDG and DvhDG for the controlled display of adhesins on the cell surface that await identification. One example is *Salmonella enterica* SiiE, an adhesin that mediates host cell attachment, is associated with a T1SS, and is retained at the outer membrane via its N-terminal moiety[66]. The retention of the adhesin on the bacterial surface is highest during host cell invasion, which may suggest a regulatory step of adhesin release[67]. Whether this step involves proteolytic processing, potentially regulated by cellular signaling, remains to be established.

## Methods

**Bacterial strains, plasmids, culture conditions, and nucleotides**
The strains, plasmids, and other resources used in the present study are listed in Supplementary Table 8. *E. coli* strains were grown aerobically at 37 °C, unless otherwise stated. Cultures were grown in

lysogeny broth (LB) (10 g/l tryptone, 5 g/l yeast extract, and 10 g/l NaCl [pH 7.5]). LB agar medium contained 1.5% (w/v) granulated agar (BD Difco). Antibiotics and inducers were used, when necessary, at the following concentrations: ampicillin, 100 µg/ml; kanamycin, 50 µg/ml; and isopropyl β-D-1-thiogalactopyranoside (IPTG), 0.5 mM. For large-scale protein expression, *E. coli* BL21DE3 (T7 Express, NEB) was grown in Terrific Broth (TB) (24 g/l yeast extract, 20 g/l tryptone, 17 mM $KH_2PO_4$, 72 mM $K_2HPO_4$, 8 ml/l glycerol) (BD). Where indicated, open reading frames were fused with functional sequences for purification or detection (msfGFP, StrepTagII, His$_6$-SUMO, or His$_6$). DNA fragments used to assemble the expression constructs were codon-optimized for *E. coli* and synthesized by Eurofins or GeneArt. Alternatively, gene synthesis and cloning of codon-optimized DNA fragments were performed by GenScript. Nucleotides were purchased from Jena Biosciences or Biolog Life Science Institute.

Recombinant proteins were expressed using either a pET21 plasmid or a modified pET28 plasmid (Novagen). The former adds a C-terminal His$_6$ affinity tag in frame, with the insert lacking a stop codon, and the latter encodes an N-terminal His$_6$-SUMO purification tag. The inserts were synthesized commercially after computational codon optimization using the preferred codon usage in *E. coli* and in-frame addition of flanking restriction sites (pET21, 5′ NdeI/3′ NotI; pET28, 5′ BamHI/3′ NotI). The inserts were introduced into multiple cloning sites of the linearized target vectors using InFusion cloning (Takara) or ligation-based methods. Plasmids were maintained and amplified in *E. coli* DH5alpha (NEB) grown in LB medium supplemented with the appropriate antibiotics. Plasmid DNA was purified using NucleoSpin Plasmid Mini kits (Macherey-Nagel), following the manufacturer's protocol. All plasmids were verified using analytical restriction digestion and Sanger sequencing (Microsynth). Site-directed mutagenesis was performed using the QuikChange II Site-Directed Mutagenesis Kit (Agilent Technologies), according to the manufacturer's protocol. Mutations were confirmed by DNA sequencing.

**Protein expression and purification.** All proteins were overexpressed in *E. coli* BL21DE3. Cultures were grown at 37 °C in Terrific Broth (TB) medium supplemented with kanamycin or ampicillin for pET28- or pET21-based constructs, respectively. At an optical density at 600 nm (OD600) of ~0.6–0.9, the temperature was reduced to 18 °C, and protein expression was induced by adding IPTG (0.5 mM). After 16 h at 18 °C, cells were harvested by centrifugation (4000 × *g*, 45 min at 4 °C), resuspended in Ni-nitrilotriacetic acid (Ni-NTA) buffer A (25 mM Tris-HCl [pH 7.5], 500 mM NaCl, 20 mM imidazole), and flash-frozen in liquid nitrogen. For soluble proteins, the cell suspensions were thawed and lysed by sonication. Cell debris was removed by centrifugation (20,000 × *g*, 45 min at 4 °C) and the cleared lysates were incubated with Ni-NTA Superflow resin (Qiagen) pre-equilibrated with Ni-NTA buffer A. The resin was washed with 20 column volumes of buffer A, followed by protein elution with five column volumes of Ni-NTA buffer B (25 mM Tris-HCl [pH 8.5], 500 mM NaCl, and 300 mM imidazole). Unless noted otherwise, the His$_6$-SUMO moiety was cleaved off overnight using the His$_6$-tagged yeast protease Ulp-1 after buffer exchange into a gel filtration buffer (25 mM Tris-HCl [pH 8.5] and 150 mM NaCl) on a fast-desalting column (Cytiva). Ulp-1, uncleaved protein, and cleaved fusion tags were removed by Ni-NTA affinity chromatography, while the target protein was recovered in the flow-through. The target proteins were subjected to size exclusion chromatography on a Superdex 200 16/600 column (GE Healthcare) preequilibrated with gel filtration buffer (25 mM Tris-HCl [pH 7.5], 150 mM NaCl). Purified proteins were concentrated on Amicon filters with an appropriate size cutoff to concentrations of >25 mg/ml, flash frozen in liquid nitrogen, and stored at −80 °C.

**Protein localization assay.** *E. coli* BL21DE3 cells expressing full-length DvhG-msfGFP, LapG-msfGFP, DvhD-msfGFP, or LapD-msfGFP were grown in LB medium (100 ml) as described above, spun down (4000 × *g*, 30 min at 4 °C), and frozen. The pellets were thawed, resuspended in fractionation buffer 1 (25 mM Tris-HCl [pH 7.5], 300 mM NaCl, and 700 µM $CaCl_2$), and lysed via sonication. Cell membranes were pelleted by centrifugation at 40,000 × *g* for 45 min at 4 °C. The supernatant (soluble fraction) was collected, and the membrane pellet was resuspended in fractionation buffer II (fractionation buffer I supplemented with 2% Triton-×100), followed by incubation under rocking for 2 h at 4 °C. The insoluble material was pelleted at 40,000 × *g* and 4 °C for 45 min, and the supernatant (membrane fraction) was collected. Soluble and membrane fractions were mixed with SDS-PAGE sample buffer (50 mM Tris-HCl pH 6.8, 2% SDS, 10% glycerol, 1% β-mercaptoethanol, 12.5 mM EDTA, 0.02 % bromophenol blue) and resolved on a 12% SDS-PAGE gel without boiling the samples. The gels were imaged on a Bio-Rad Imager using the appropriate fluorescence channel to detect GFP (msfGFP; "in-gel fluorescence").

**Proteolysis assay.** DNA fragments from DvhA (corresponding to residues 79–129 and residues 1–129) or Dvu1545 (corresponding to residues 59–111 and 1–111) were synthesized and cloned into the pCleevR plasmid described by Cooley et al.[19] via InFusion cloning (Takara). LapA 81–131 in the pCleveR plasmid has been described before[19]. Upon expression following the protocol described above, a protein is made in which the inserted sequence is flanked by an N-terminal His$_6$-SUMO tag and C-terminal msfGFP (Supplementary Table 9). The proteins were purified using Ni-NTA affinity chromatography, followed by size-exclusion chromatography. Full-length DvhG-StrepTagII in the plasmid pCDF-Duet or His$_6$-SUMO-LapG in the plasmid pET28 were expressed in *E. coli* BL21DE3, as described above. Cells were collected by centrifugation at 4000 × *g* at 4 °C for 45 min and resuspended in 25 mM Tris-HCl (pH 7.5), 300 mM NaCl, and 5 mM $CaCl_2$. Cells were lysed by sonication, followed by solubilization for 1.5 h in buffer supplemented with 2% (v/v) Triton-×100. The insoluble material was removed by centrifugation at 65,000 × *g* and 4 °C for 45 min. The supernatants were normalized to total protein and used for subsequent cleavage assays. Lysates containing DvhG or LapG were incubated with 20 mM EGTA, as indicated. Purified protease reporters (30 µM final concentration) were added to the lysates (90 µl) containing DvhG or LapG or serial dilutions thereof for dose-response curves, followed by incubation for 60 min at room temperature. Samples (without boiling) were analyzed using 12% SDS-PAGE gels and imaged on a Bio-Rad Imager using the setup for in-gel fluorescence (GFP/Alexa 488) and regular light (Coomassie) to image the molecular weight ladder. Both gel images are overlaid. Fluorescence intensities were quantified using ImageJ/Fiji and plotted as percent cleaved[68]. StrepII-tagged DvhG was detected by Western blotting using a Strep•Tag II antibody horseradish peroxidase conjugate (Sigma-Aldrich, catalog number 71591) at a 1:4000 dilution in PBS/0.1% Tween20 for detection.

**Nano differential scanning fluorimetry (nanoDSF).** HD-GYP and PAS/HD-GYP domains of DvhD were purified as described above. Proteins (257 µM) were mixed with c-di-GMP (0.070–2 mM). For HD-GYP point mutants and for experiments concerning c-di-AMP, 3′3′-cGAMP, pGpG, and pApG, 3 mM of ligand was used. Fluorescence signals were measured at 330 and 350 nm using the Nanotemper Prometheus NT.48, along a temperature gradient from 25 to 95 °C, with the temperature increasing at 1 °C/min and fluorescence measurements taken every 0.05 °C. Unfolding curves were generated using MoltenProt software fitted with an equilibrium two-state model with an 8 °C temperature range for baseline separation and a baseline separation factor greater than 0.5[69].

The melting temperature of the PAS/HD-GYP protein was determined by evaluating the maximum fluorescence value of the first derivative of the ratio of the fluorescence signals at 330 and 350 nm.

For the isolated HD-GYP domain, the maximum fluorescence value of the first derivative of the fluorescence signal at 350 nm was used. Ligand-dependent melting temperature shifts are the differences in averages from three independent measurements in the presence or absence of a ligand. The standard deviation of the measurements was used to calculate the errors, which were then combined using an error propagation formula.

**Size-exclusion chromatography-coupled multi-angle light scattering (SEC-MALS).** For SEC-MALS experiments, proteins (40 μl at 1–10 mg/ml) were injected onto a Superdex 200 10/300 column (GE Life Sciences) equilibrated in gel filtration buffer at 20 °C[70]. For experiments conducted in the presence of c-di-GMP, nucleotides were added to each protein sample (at a final concentration of 500 μM). Where indicated, the SEC-MALS buffer was supplemented with c-di-GMP (20 μM). The gel filtration setup was coupled to a UV detector (Agilent 1260 Infinity II VWD), 3-angle light scattering detector (mini-Dawn; Wyatt Technologies), and refractive index detector (Optilab; Wyatt Technology). Data were collected every second at a flow rate of 1.0 ml/min. Data analysis was performed using ASTRA software (Wyatt Technology). Monomeric BSA (Sigma) was used to normalize the light-scattering detectors and for data quality control.

**Isothermal titration calorimetry (ITC).** The ligands (c-di-GMP, pGpG, 3'3'-cGAMP, or c-di-AMP) were dissolved in gel filtration buffer and titrated into a solution of the purified PAS/HD-GYP fragment of DvhD using a MicroCal PEAQ-ITC instrument (Malvern Panalytical). Injections of ligand (first injection: 0.5 μL; subsequent injections: 4 μL; spacing 150 s) were performed at 21 °C. Data were fitted and analyzed using PEAQ software (Malvern Panalytical). A single-site binding model fits the data most optimally, and the apparent binding characteristics were calculated based on this fit. Data, including experimental details, are shown in Supplementary Table 5.

**Protein structure prediction.** We employed AlphaFold2 and Colab-Fold as well as AlphaFold3 to predict protein structures using default parameters[71–73]. Five models were generated and ranked for each prediction. For DvhD, a dimeric quaternary structure was assumed, and DvhG was modeled as a monomer. The top-ranked models were relaxed by molecular dynamics within ColabFold. Confidence scores (i.e., pLDDT, pTM, ipTM) for the models shown are summarized in Supplementary Table 1 and Supplementary Data 2. Signal peptides and transmembrane domains were predicted using SignalP-6.0 and TMHMM-2.0, respectively[64,74].

**Structure determination via X-ray crystallography**
**DvhD^dCache.** The purified dCache domain of DvhD (residues 57–327 of *D. vulgaris* Hildenborough DVU1020) was produced as a selenomethionine-derivatized protein. Briefly, the expression plasmid was transformed into T7 Express *E. coli* BL21DE3 cells (NEB) and plated on LB agar plates supplemented with the appropriate antibiotics for selection. After overnight incubation, liquid medium (200 ml LB/antibiotic) was inoculated and shaken for 4 h at 37 °C (OD600 > 1). The starter culture was used to inoculate 4 l of LB/antibiotic media, followed by shaking at 37 °C until an OD600 of 0.7 was reached. The cells were centrifuged at 4000 × *g* for 25 min at 18 °C, washed once with M9 salt medium, and centrifuged again. The cell pellets were resuspended in 1 l of minimal medium consisting of M9 salts, antibiotics, 40 mg of each amino acid except methionine, 40 mg selenomethionine, 2 mM magnesium sulfate, 25 μg iron sulfate, and 0.4% glucose. The cells were shaken at 37 °C for one hour before the temperature was reduced to 18 °C and protein expression was induced with 1 mM IPTG. After 12–16 h, the cells were harvested, and protein purification was performed as described above. The purified, selenomethionine-derivatized dCache domain of DvhD was crystalized in 0.2 M

ammonium acetate, 0.1 M Bis-Tris pH 6, 27% polyethylene glycol 3350 (Hampton Research), and 20% xylitol in hanging-drop vapor diffusion at 20 °C.

**DvhD^HD-GYP with c-di-GMP.** The purified HD-GYP domain of DvhD (residues 491-701 of DVU1020) was set up in a sitting-drop vapor-diffusion INDEX screen (Hampton Research) at 0.572 mM (13 mg/ml) with c-di-GMP (Jena Biosciences) at a final concentration of 1 mM. The protein crystallized in 0.2 M ammonium acetate, 0.1 M Bis-Tris pH 6.5, 25% (w/v) polyethylene glycol 3350 at 20 °C. Crystals were flash frozen in liquid nitrogen after being soaked in 0.2 M ammonium acetate, 0.1 M Bis-Tris pH 6.5, 25% (w/v) polyethylene glycol 3350, and 20% xylitol.

**DvhD^PAS/HD-GYP.** The purified PAS/HD-GYP fragment of DvhD (residues 374–701 of DVU1020) crystallized in 0.1 M Hepes (pH 7.5), 10% (w/v) polyethylene glycol 6000, and 5% (v/v) MPD in a sitting drop vapor diffusion set up at 20 °C.

All crystals were flash-frozen in liquid nitrogen after soaking in LV cryo-oil (MiTeGen). Diffraction data were collected at 100 K using the PETRAIII beamline P11 at the Deutsches Elektronen-Synchrotron (DESY) (Hamburg, Germany). Data were processed using XDS[75]. The initial phase information was obtained either by single-wavelength anomalous diffraction (DvhD^dCache) using the PHENIX autosol module or by molecular replacement using PHENIX phaser module (DvhD^HD-GYP and DvhD^PAS/HD-GYP) in the PHENIX software package, with a search model derived from AlphaFold2 models based on the DvhD primary sequence[76–78]. Models were iteratively refined using the software COOT[79] and PHENIX.refine[76,80]. Data acquisition and refinement statistics are summarized in Supplementary Table 4.

**High-performance liquid chromatography (HPLC).** Purified proteins (20 μM) were mixed with nucleotides (200 μM) in a total reaction volume of 60 μl in 25 mM Tris-HCl (pH 7.5) and 500 mM NaCl. The samples were incubated at 20 °C for 90 min, followed by heat denaturation at 95 °C for 20 min. Denatured proteins were removed by centrifugation at 21,000 × *g* for 10 min and filtration of the supernatants through a 0.22 μm-spin filter. Each sample (20 μl) was injected onto a C18 reverse-phase column (Gemini 3 μm C18, 110 Å, 150 × 4.6 mm; Phenomenex) equilibrated with 100 mM $KH_2PO_4$, followed by gradient elution from 100 mM $KH_2PO_4$ to 70% $KH_2PO_4$ and 30% methanol. During the gradient, the UV signal at 260 nm was recorded (Agilent 1260 Infinity II VWD). Purified standards were analyzed to determine the elution times of specific pure nucleotides for comparison.

**Evolutionary conservation mapping.** DvhD homologs were identified via a BLASTp search against the "refseq_selected" database with maximal target sequence number ("max target sequences") set to 1000[81,82]. The results were further filtered to include only sequences with a minimal query coverage of 85% and an E-value cutoff of $1.0e^{-50}$, resulting in the selection of 78 high-confidence, representative homologs with an alignment score greater or equal to 200 that cover all domains of DvhD. The webFlaGs server was used to inspect the genomic neighborhood of the selected 78 entries, spanning 12 genes flanking the DvhD-encoding genes (Supplementary Data 1)[59]. From this analysis, the corresponding DvhG-homologous sequences were identified. Multi-sequence alignments were generated using Clustal Omega with the DvhG and DvhD sequences as the input and applying default parameters[83,84]. The resulting alignment file in conjunction with an AF2-based model of a DvhD dimer or a DvhG monomer were used as inputs for Consurf (Consurf Web Server) with otherwise default parameters[65,85,86]. The structural models with conservation scores mapped onto the surface and colored using Consurf default values were visualized in ChimeraX[87].

For phylogenetic mapping, DvhD and DvhG (BTLCP domain only) orthologs were identified via a BLASTp search against a non-redundant protein sequence database and applying a sequence coverage cutoff of 80%. Phylogenetic trees were generated with PhyloT (v2) based on the NCBI taxonomy identifiers for organisms containing both DvhD and DvhG orthologs and visualized using iTol (v6)[88].

## Reporting summary
Further information on research design is available in the Nature Portfolio Reporting Summary linked to this article.

## Data availability
Unless otherwise stated, all data supporting the results of this study can be found in the article, supplementary, and source data files. The coordinates and structure factors have been deposited in the Protein Data Bank under accession codes 9RBZ (DvhD HD-GYP domain•c-di-GMP), 9RC0 (DvhD PAS/HD-GYP domains), and 9RC2 (DvhD periplasmic domain)[89]. Crystallographic and predicted structural models have been deposited at the figshare repository under the link [https://doi.org/10.6084/m9.figshare.31136614]. The previously published coordinates for a *P. fluorescens* LapD•LapG complex, *L. pneumophila* LapG, *Pm*GH•c-di-GMP, KinD[dCache], SpdE[dCache], PctA[dCache], PctD[dCache], and McpH[dCache] referred to in this study are available at the PDB under the accession codes 4U65, 4FGO, 4MDZ, 4JGR, 7K5N, 5T7M, 7PRQ, and 8BMV, respectively. Source data are provided with this paper.

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

## Acknowledgements

We acknowledge technical support from the SPC facility at EMBL Hamburg. We further acknowledge DESY (Hamburg, Germany), a member of the Helmholtz Association HGF, for the provision of experimental facilities. Parts of this research were carried out at PETRA III, and we would like to thank Johanna Hakapää and the entire P11 staff for their assistance in using beamline P11. This work was supported by the National Institutes of Health (1R01AI168017 to G.A.O.) and the Deutsche Forschungsgemeinschaft (DFG, German Research Foundation) (564279481-SPP2474 to H.S.).

## Author contributions

M.E.F., G.A.O., and H.S. conceived and planned the experiments. M.E.F., J.D.L., S.M., M.J.G.G., and H.S. carried out the experiments. M.E.F. and M.J.G.G. contributed to sample preparation. M.E.F., A.A.K., G.A.O., and H.S. contributed to the interpretation of the results. M.E.F. and H.S. took the lead in writing the manuscript. All authors provided critical feedback and helped shape the research, analysis, and manuscript.

## Funding

## Competing interests

The authors declare no competing interest.
