## [Transparent Peer Review file · Nature Communications]

Structural analyses uncover protease-adhesin interactions and c-di-GMP receptor regulation in sulfate-reducing bacteria

Corresponding Author: Professor Holger Sondermann

Version 0:

Reviewer comments:

Reviewer #1

(Remarks to the Author)

Summary and General Assessment

This manuscript examines a proposed *Desulfovibrio vulgaris* system that regulates cleavage of the RTX adhesins DvhA and DVU1545, focusing on the protease DvhG and the putative c-di-GMP receptor DvhD. The paper brings together a diverse set of approaches, including sequence and structural alignments, AlphaFold2 modeling, proteolysis assays, localization experiments, in-gel fluorescence, mutagenesis, X-ray crystallography, SEC-MALS, and nanoDSF, to investigate these proteins. The topic is timely and relevant, as it seeks to extend paradigms of c-di-GMP signaling into sulfate-reducing bacteria where some components resemble canonical Lap system proteins but others lack clear homology.

While the manuscript presents a broad methodological effort and generates some intriguing observations, particularly regarding the possibility of a noncanonical “bridging” c-di-GMP binding site in DvhD, the evidence as currently presented is not yet cohesive. In its current form, the study reads as two independent stories that are partially connected, and some of the central claims rely heavily on predictive modeling modest experimental validation. Additional work, such as quantitative biophysical assays and integrative modeling of DvhD–DvhG interactions, as well as reorganization of the story will be needed for publication in a high-impact journal like Nat Comm.

Major Comments

The manuscript builds on the canonical Lap system in *Pseudomonas fluorescens* to investigate orthologous components in *Desulfovibrio vulgaris*, with an emphasis on how the protease DvhG recognizes and cleaves RTX adhesins and how the receptor DvhD senses c-di-GMP to regulate this process. The paper combines structural biology, biochemistry, and modeling, and the study is technically ambitious. However, the conclusions rely heavily on AlphaFold2 predictions to support structural hypotheses. Given the recent availability and superior accuracy of AlphaFold3, especially for modeling relatively large protein–protein complexes, it is essential that all analyses currently based on AlphaFold2 be repeated or validated using AlphaFold3. This would provide much stronger confidence in the proposed interaction models and substrate recognition mechanisms. In addition, all AlphaFold-based results must be accompanied by confidence metrics such as pLDDT and PAE values to improve transparency and reproducibility.

A central weakness of the manuscript is the limited integration between the two parallel stories: the biochemical characterization of DvhG–adhesin interactions and the structural analysis of DvhD. While the paper proposes that DvhD regulates DvhG, no structural or modeling evidence is provided to link these proteins mechanistically. Given the otherwise extensive use of AlphaFold predictions, the absence of attempts to model DvhD–DvhG complexes and of experimental work is conspicuous. I recommend that the paper includes AlphaFold3 modeling of the DvhD–DvhG complex. Without such integration, the manuscript reads as two largely disconnected studies, rather than as a cohesive regulatory model. Several other technical issues should also be addressed. The proteolysis assays, while clearly described, would benefit from quantitative analysis of cleavage efficiency using image-based densitometry. Presenting percentages of substrate cleaved, along with basic descriptive statistics (e.g., mean values & standard deviations), would allow more precise comparison across experimental conditions.

The crystallographic data are generally acceptable, but the relatively high R-work and R-free values for the periplasmic dCache and HD-GYP domain structures should be explicitly acknowledged. In addition, the deposited coordinates are

confusingly labeled: all three PDB entries appear under the same name, "DvhD HD-GYP domain, c-di-GMP-bound," even though they represent distinct constructs or conditions.

The identification of a noncanonical "bridging" c-di-GMP site in DvhD is novel and of clear interest to the field. However, the conclusions about ligand specificity currently rely almost entirely on nanoDSF stabilization assays, which are indirect (No statistical significance was indicated). These data suggest modest stabilization by alternative ligands such as 3'3'-cGAMP, c-di-AMP, and pGpG, but do not establish quantitative binding affinities or stoichiometries. I recommend that the manuscript includes additional binding experiments, such as ITC which can measure K_d , stoichiometries (N), and other thermodynamic parameters. Such data would build on their prior ITC work with c-di-GMP ($N \approx 1.4$), which suggested mixed occupancy of canonical and bridging sites, and would clarify whether alternative ligands engage the same sites or use distinct binding modes. ITC experiments (ideally in 2 or 3 replicates) with targeted site mutants could further discriminate between canonical and bridging site interactions. Most importantly, these additional experiments would address the key question of physiological relevance. At present, it is not clear whether the modest stabilization observed with c-di-AMP, 3'3'-cGAMP, or pGpG reflects biologically meaningful binding *in vivo*. Quantitative affinity and stoichiometry measurements will be helpful to distinguish these possibilities and help frame a more coherent regulatory model.

The manuscript is focused on description and discussion of the authors' previous work on *P. fluorescens* and *D. vulgaris* with some mention of the *V. cholerae* systems. Given the Lap system is well conserved in Gram-negative bacteria and is of broad interest, the manuscript should include more rigorous discussion and citations on other works in the field such as the Antarctic bacterium *Marinomonas primoryensis*, and slightly different adhesin retention systems such as those in *Salmonella enterica* (e.g. SiiE).

Other Minor comments to be addressed:

There are some inconsistencies in figures and text. Better proofreading will be necessary for the revised manuscript.

For example,

- In Figure 5, the hydrogen bond is attributed to residue E448 in the text, but the figure labels E668.
- Ensure consistency for "Phosphodiester bond" (e.g. line 309) and "phosphodiesterase bonds" (line 313).
- the phrase following the statement on lines 167-169 describing the second panel of Figure 2B should be revised for better accuracy. "A substrate analog with a di-arginine instead of a di-alanine sequence in the DvhA '[T/A/P]AAG' motif failed to be cleaved in the presence of either protease, confirming the di-alanine sequence as the scissile bond." The figure actually showed relatively faint bands at ~ 26 kDa for all conditions except for the first column (control group with no protease). The PRRG mutation may be more resistant to proteolysis by DvhG and LapG than the PAAG motif.
- In Fig. 3C, the top panels and bottom panels represent different constructs with the experiments done with the retention module in the bottom panels. Why do the uncleaved proteins in both the top and bottoms appear to have the same size despite the constructs in the bottom panels have the additional retention domain? The full sequence of the constructs used in the proteolysis assay can be included in the supplemental document.
- The nucleotide c-di-AMP, which seems to have modest stabilization of DvhD per Figure 7, is known for signalling in bacterial cells, but the biological implication was not discussed. The possible biological significance for the alternative ligands for DvhD, such as linear di-GMP (pGpG), GMP, and GTP could be better discussed.
- Fig. 5C, not useful to superimpose... one can already compare 5B above with 5C
- Fig. 6C does not explain the dotted versus continuous curves (the light-scattering and refractive index signal) as clearly as Figure S8.
- It would be helpful to label temperatures of the peaks. (Fig. 7A and Fig. S6)
- Statistical analyses, such as ANOVA and t-tests, maybe helpful for nanoDSF results shown in Figure 7.
- The gel Figure S1C should show the top 3 bands on the protein marker. The size of these markers supports the size of the bands given asterisks (** and *** in particular). This would strengthen claim that the fluorescent bands have twice the size of the monomers and indicate possible dimerization.

Reviewer #2

(Remarks to the Author)

Reviewer #3

(Remarks to the Author)

In essence, this manuscript describes a comparative study of one system with another that has been more extensively

investigated. It focuses on the deltaproteobacterial Dvh system, comparing it with the gammaproteobacterial Lap system. In their cognate hosts, both systems are responsive to the second messenger c-di-GMP to regulate biofilm formation and bacterial dispersal. Their components are functionally analogous but structurally distinct. In both systems, a protease processes adhesins which are critical for cell attachment. The results in this manuscript extended the functional analogy of Dvh to Lap at the mechanistic/biochemical level and explored the implications of their structural divergence in maintaining the functionality of the former.

The authors proposed a model wherein the Dvh system functions similarly as the Lap system. When c-di-GMP concentration is high, the transmembrane DvhD (LapD counterpart) binds c-di-GMP through its cytoplasmic portion. This binary complex sequesters the protease DvhG (LapG equivalent, periplasmic), preventing it from cleaving and freeing the adhesins from the cell surface. Lack of proteolytic processing allows the adhesins, DvhE and DVU1545 (LapA equivalents), to be anchored to the cell surface through a retention (R) domain located in the periplasm. In this state, these adhesins are attached to the cell surface to facilitate and promote biofilm formation. When c-di-GMP concentration is low, the apo-DvhD releases DvhG, which then cleaves off the R domain of the adhesins. The anchorless adhesins slip out of cells to allow bacterial dispersal from biofilms.

The authors took a multiple disciplinary approach in this study. There are two parts to this manuscript, both of which used AlphaFold modeling to demonstrate the structural differences and similarities of Dvh components with their Lap counterparts. In the first part, they used *E. coli*, a gammaproteobacterium, to express proteases of both systems. Fractionation was used to investigate protein localization and whole cell lysates expressing the proteases for proteolytic processing of their respective substrates. Although these experiments are not perfectly clean, appropriate controls were included to ensure the conclusions are solid. The second part of the manuscript focused on DvhD, the transmembrane c-di-GMP receptor of the Dvh system. They purified three different parts of DvhD: 1) its periplasmic dCache-like domain, 2) its entire cytoplasmic part (PAS + HD-GYP domains) and 3) the HD-GYP domain by itself. The last two truncated variants were shown to bind c-di-GMP with the longer one being a dimer and the shorter one being a monomer that dimerizes in the presence of c-di-GMP. Moreover, they determined the X-ray crystal structures of these three with the surprising discovery of a third c-di-GMP molecule binding at the interface of the HD-GYP dimer.

The overall quality of the work and the manuscript is high, with substantial new experimental results and insights. The proposed model is consistent with the authors' observations and appears highly plausible. One major issue is that the authors have attempted to tackle nearly all aspects of their model, some of which are more strongly substantiated than others.

Major points:

- 1) There is no direct evidence or information for the periplasmic interaction between DvhD and DvhG and the effect of c-di-GMP on this interaction. Bacterial two-hybrid system may be considered, especially because c-di-GMP levels in *E. coli* could be artificially manipulated.
- 2) In the context of c-di-GMP binding and its effect on DvhD, is there an experimental structure of PAS/HD-GYP in complex with c-di-GMP? Was there an attempt to soak the PAS/HD-GYP crystal with c-di-GMP or crystallization in the presence of c-di-GMP? This seems logical as the PAS/HD-GYP fragment clearly binds c-di-GMP from the current and past results. It would be better to compare the apo form of PAS/HD-GYP with its c-di-GMP complex than with the binary complex of HD-GYP and c-di-GMP.
- 3) In light of the unexpected c-di-GMP binding at the interface in the crystal structure of the HD-GYP domain, it would be helpful to revisit the binding of c-di-GMP to this domain and to the PAS/HD-GYP fragment directly with more analytic techniques (than thermostability). More nuanced experimental design and data analysis may reveal multiple binding affinities and more precise stoichiometry. In this context, the authors' previous publication in PNAS showed that one molecule of PAS/HD-GYP binds 1.4 molecules of c-di-GMP, consistent with 3 ligands for one dimer. Related to point #2) above, the biophysical data (ITC and thermostability) suggest possibly a higher resolution for the crystal structure of PAS/HD-GYP if complexed with c-di-GMP.
- 4) There is too much focus on the Lap system in the writing. I suggest to focus more on Dvh with Lap for comparison and interpretation purposes after presenting results. A couple of examples below.
 - a. The first two paragraphs under the subheading "Adhesion retention domain..." could be switched. Consider using the Lap system for comparison and data interpretation instead of a lead-in for the Dvh experiments.
 - b. Sentence starting on L242 would be a better opening for the relevant paragraph. The remaining part of the paragraph could be re-structure to state something like: "Unlike LapD, which has xxx domain for c-di-GMP binding, DvhD contains an HD-GYP domain with the potential to bind c-di-GMP. We investigated.....". In this context, the structure of the periplasmic domain seems out of place as its proposed interaction with DvhG is not investigated in anyway.
- 5) It seems that the biochemical/biophysical studies of DvhD should precede the structural studies in the organization of the manuscript. These include c-di-GMP binding and analysis of oligomeric states.

Minor points:

- 1) More informative summary paragraphs or concluding sentences would be helpful.
- 2) There should be explicit indications of *D. vulgaris* as a deltaproteobacterium and all others are gammas in the writing. Speculate a little on whether the cases here represent structural divergence or functional convergence in evolution.
- 3) It is unclear why *E. coli* is used here instead of *P. fluorescens* which allowed a successful reconstitution of the Dvh system for functional analysis.

(Remarks to the Author)

The manuscript by Font and colleagues describes a structural and biochemical study of the DvhD/G/A system, responsible for the synthesis and exposition of adhesins in *Desulfovibrio vulgaris* Hildenborough. This system is present in Sulfate-reducing bacteria and relate to the LapG/D/A system that exists in Gammaproteobacteria.

The authors have assemble a remarkable number of evidence, based on structural modelling, crystal structures, clever biochemical assays and mutagenesis to illuminate the mechanism of regulation of adhesin fixation and release at the outer membrane of *Desulfovibrio vulgaris* Hildenborough. In particular, this work provides a detailed mechanism and a structural basis of the cleavage activity and activation of DvhG as well as of ci-diGMP sensing by the DvhD membrane protein

I have found the evidence compelling, the structural work excellent except for one crystallographic structure that could be improved (see below) and the model strongly supported by the data.

One putative drawback is that no experiment has been done in vivo to see if the mechanism described exist in the *Desulfovibrio vulgaris* Hildenborough. It would have been nice if some mutants had been introduced in vivo to inspect the consequences on biofilm formation in this bacteria or alternatively in the model system *Pseudomonas fluorescens*.

Another issue is the structure of the DvhD PAS/HD-GYP domain solved at 3.4Å (Table S1). There seem to be 1% Ramachandran outlier which is rare. Moreover, the R-free and R-factor are quite high as well, with 0.33/0.31 suggesting that more work can be done on the model. Can the author provide a model and structure factor of this structure so that we can verify that this is a real outlier ? Alternatively a bit more work on the structure refinement might help so that we can be sure that the model is fully refined.

In some parts, in particular the introduction, the paper is written for specialist of these systems and thus, except for a few aficionados, it might not be comprehensible for some non experts of the field.

I have tried to include some examples below. Many of the domain names are not defined. Including these definition would really help.

Moreover a clear definition of the representation of this system in the various bacteria would be a plus. The supplementary dataset S1 (presented in discussion) does not really help. Can the author provide a simple bacterial phylogenetic tree with the order/genus that encode for these systems ? OR something more comprehensible ?

Otherwise I have few minor comments in order to improve an already very good manuscript.

Details of the minor issues

L101 L102 what are the PAS and HD-GYP domains ? Are these really important for understanding the paper ? if yes then they should be defined. What does PAS or HD-GYP stand for ?

L124

L141 « correlating with the lack of a lapD-like gene » I thought both the drawing (Fig1) and introduction suggest that DvhD is an homologue of LapD ? (for instance L98 ?). A clearer explanation arrives only L244 with « A recent study has shown that DvhD, enables c-di-GMP-dependent control over DvhG in a *P. fluorescens* strain lacking LapD and LapG, supporting that DvhD could be the functional homolog of LapD in SRB 25 . However, DvhD bears no sequence or domain resemblance to LapD, leaving the open question of how DvhD senses c-di-GMP levels and transduces this signal to regulate adhesion and biofilm formation in *D. vulgaris* ».

Can the author consider reformulating these parts so that it is clearer from the introduction ?

L149. « comprises a predicted (targeting) helix with low sequence identity, » is this supposed to mean low sequence conservation ?

L271; dCache. Please define what dCache domains are and what they are used for in bacteria. Are these specific to bacteria ?

Figure 2 would have been easier to follow with a schematic description of the adhesin domains and boundaries

Figure 6 juxtamembrane should be juxtamembrane

Reviewer #5

(Remarks to the Author)

This manuscript by Font et al. focuses on molecular mechanism of adhesion/dispersion and biofilm formation in a sulfate-reducing bacterium. Structural analysis including three X-ray crystal structures and Alpha Fold2 prediction and detailed biochemical analysis provide a deep insight about the mechanism involving key proteins, DvhD, DvhG, and DvhA. It is interesting that the signal by c-di-GMP binding to DvhD is transduced from cytosol to periplasm in outward direction over the inner membrane in the Dvh systems. It is expected that the comprehensive study attracts wide attention of not only specialists of the related fields but also the general readership of Nature Communications. Potentially, this work is valuable

to be published in this journal. However, the explanation or description was insufficient in some parts, and several important points should be reconsidered before publication.

Major concerns

1. Lines 454

The titrated site number ($n = 1.4$ per protein mol) calculated from ITC measurement in a previous study (ref. 25) is supportive for Author conclusion that 3 mol of c-di-GMP is bound to HD-GYP domain. K_d value, another parameter obtained from the ITC experiment, was single value (322 nM, ref. 25). This means that the three potential binding sites for c-di-GMP have nearly identical affinity and that single transition for the c-di-GMP binding is observed in the ITC experiment. However, I guess that among three potential binding sites observed in X-ray crystal structure of the complex, two canonical sites for c-di-GMP should have clearly higher affinity than that of a single bridging site, because a dense network of hydrogen bonds and electrostatic interactions are formed only for the canonical sites. If that is the case, two titration steps would be measurable. Is this inconsistent with Author conclusion? According to Fig. 5D, the number and degree of conservation of the interacting amino acid residues at the bridging site (#, 3 conserved/4 positions) are less than those at the canonical sites (§, 7 conserved/10 positions). Alternatively, I suspect that the c-di-GMP binding at the bridging site is low affinity site and an artifact under crystallization conditions in which 1 mM of high concentration of c-di-GMP (3000 folds of K_d) is contained.

2. When nevertheless Authors claim that one more c-di-GMP is really bound at the bridging site in the sulfate-reducing bacteria, Authors need to discuss physiological significance of the binding of three molecules of c-di-GMP in the DvhD signal sensing mechanism in comparison with LapD dimer in which the binding of two molecules of c-di-GMP is required.

In addition, this study showed that DvhG is anchored on the inner membrane. Does this feature also have any physiological meaning in the Dvh system?

3. Figure 6A, Lines 339-340

This study suggests that the large conformational change of stalk helices of DvhD dimer is induced upon the binding of c-di-GMP by comparing N-terminal position of the stalk helices between free PAS/HD-GYP domain and c-di-GMP-bound HD-GYP domain. However, the conformation change upon the ligand binding is estimated using two constructs having different domain architectures. Is it OK? It is likely that the PAS region of the free PAS/HD-GYP domain interact each other in the homodimer. Therefore, lack of PAS domain may affect the difference of conformation in c-di-GMP-bound HD-GYP domain. If the conformational change of the stalk helices is really induced by c-di-GMP binding, the remaining N-terminal parts including dCACHE domain, trans-membrane helices, and PAS domain will be separated from the counter part?

In addition, in crystal lattice, the N-terminal region of c-di-GMP-bound HD-GYP domain may be also affected by the adjacent molecules. Does the crystal packing affect the conformation of stalk helices? These points should be mentioned in the manuscript. Although X-ray crystal structure of DvhD PAS/HD-GYP domain is low resolution, the structures of the binding site of c-di-GMP and the surrounding residues in the c-di-GMP-bound HD-GYP domain will provide the information for structural change upon the c-di-GMP binding, expecting the details of initial trigger for structural change. Did Authors check it?

Minor concerns

1. To clearly express distinct molecular architectures between DvhD and LapD, LapD structure should be also shown in the manuscript. The overall dimer model of LapD predicted by Alpha Fold2 is useful for the comparison with DvhD. Authors are able to add the comparison to Figure 4 or as new supplemental data.

2. Fig. 4C legend

A sentence "Dashed and solid curves show the light-scattering and refractive index signal from a size exclusion-coupled ..." means "Dashed and solid curves show the light-scattering and refractive index signals, respectively, from a size exclusion-coupled ..."?

For easier understanding SEC-MALS data for most of readers, Authors also cite an appropriate reference for the SEC-MALS experiments here.

3. Several residues that are not found in Figures were cited in the main text: E448 (line 307; E668?), Y562A (line 396; Y652A?), etc. If they are mistakes for residue numbers, please fix them. Or the positions of those residues were not shown in Figures. The positions of the cited residues must be indicated in the corresponding figures. In addition, thermal stability data about M646D was not found in Figs. 7B and 7C (line 397). Please check again over the entire manuscript.

4. Many figures and tables lack explanation for contents in their legends. Authors should provide self-explanatory legends. For example, Table S2 has no legend. To understand this table, most of readers require explanation of many abbreviations contained in the table: "Pred." "SP", "GLOB", "S", and "O". Reference for DeepTMHMM should be also cited here. Authors should again check whether all figures and tables in the main manuscript and extended data have self-explanatory legends and whether abbreviations are defined here or elsewhere.

5. Line 493

b-D-1-493 thiogalactopyranoside - β -D-1-493 thiogalactopyranoside (change to symbol font)

Version 1:

Reviewer comments:

Reviewer #1

(Remarks to the Author)

The authors have addressed my comments and concerns with extensive textual revision and additional experiments. In my opinion, the revision has significantly improved the manuscript. I have no additional criticism.

Reviewer #2

(Remarks to the Author)

Reviewer #3

(Remarks to the Author)

The revisions have successfully addressed and/or responded to all of my concerns. I believe the manuscript is stronger and I have no further comments.

Reviewer #4

(Remarks to the Author)

The authors have successfully addressed all my questions and have modified the manuscript accordingly. I have no further questions on this article.

Reviewer #5

(Remarks to the Author)

I have carefully checked point-by-point responses and reviewed the revised manuscript. Author has appropriately addressed all of my concerns as well as those of the other reviewers. I recommend the manuscript for acceptance.

Dear Editor,

We would like to thank you and the five reviewers for their careful evaluation of our manuscript. We are grateful for the constructive comments that were raised during the review process and used these to improve our manuscript. Please see our point-by-point response below.

Several reviewers noted the usage of AlphaFold predications in our study, suggesting that models remained untested. We would like to point out that most models, especially those that bear mechanistic insight and provide the basis of testable hypotheses, were validated by experimental approaches. In particular, we

- a. modeled DvhG revealing a conserved BTLCP fold of the catalytic domain. This model was validated by protease assays including mutants at the active and calcium binding sites that were designed based on the AlphaFold models;
- b. modeled LapG/DvhG•substrate complexes. Complexes involving LapG confirmed previous results and were used as proof of concept. Complexes involving DvhG indicated a more extensive interface between the protease and its adhesin substrates, suggesting a notable difference to LapG and its targets. The impact of this additional interface was confirmed by proteolysis assays including interface mutants informed by the AlphaFold models. We also showed similar results for both adhesins that are processed by DvhG, further supporting the modeling;
- c. modeled the full-length DvhD transmembrane protein. The model suggested a periplasmic dCache domain that could not be readily identified by sequence comparisons. We confirmed this result by determining the experimental structure of the periplasmic domain, albeit at low resolution. Likewise, the AlphaFold model suggested dimeric, cytoplasmic PAS/HD-GYP domains, which we confirmed by experimental structure determination and light scattering. Here, experimental structures were important because of the moderate confidence scores for the AlphaFold models of the full-length DvhD dimer.

Exceptions where AlphaFold models were introduced without further validation are the presentation of the topology and modular nature of the adhesin proteins as well as the DvhDG complex showing an interaction between the two. The adhesin models served illustrative purposes and pinpointed the position of the so-called targeting helix in the adhesin's periplasmic domain, information that motivated DvhG-substrate modeling. The DvhD-DvhG complex model provides a structural model that we hope to test in the future. Currently, this is challenging since binding studies are negative or inclusive. See below our point-by-point response for more extended explanations.

Best regards,
Holger Sondermann (on behalf of the authors)

Point-by-point response to reviewer comments:

Reviewer #1 (Remarks to the Author):

Summary and General Assessment. This manuscript examines a proposed *Desulfovibrio vulgaris* system that regulates cleavage of the RTX adhesins DvhA and DVU1545, focusing on the protease DvhG and the putative c-di-GMP receptor DvhD. The paper brings together a diverse set of approaches, including sequence and structural alignments, AlphaFold2 modeling, proteolysis assays, localization experiments, in-gel fluorescence, mutagenesis, X-ray crystallography, SEC-MALS, and nanoDSF, to investigate these proteins. The topic is timely and

relevant, as it seeks to extend paradigms of c-di-GMP signaling into sulfate-reducing bacteria where some components resemble canonical Lap system proteins but others lack clear homology. While the manuscript presents a broad methodological effort and generates some intriguing observations, particularly regarding the possibility of a noncanonical “bridging” c-di-GMP binding site in DvhD, the evidence as currently presented is not yet cohesive. In its current form, the study reads as two independent stories that are partially connected, and some of the central claims rely heavily on predictive modeling modest experimental validation. Additional work, such as quantitative biophysical assays and integrative modeling of DvhD–DvhG interactions, as well as reorganization of the story will be needed for publication in a high-impact journal like Nat Comm.

Thank you for the detailed, constructive assessment of our work. We view the strength of our study in providing an integrated, comprehensive analysis of the DvhDG system. In the revised manuscript, we have closed the gaps this reviewer pointed out by addressing the specific comments – see details below.

Major Comments

1. The manuscript builds on the canonical Lap system in *Pseudomonas fluorescens* to investigate orthologous components in *Desulfovibrio vulgaris*, with an emphasis on how the protease DvhG recognizes and cleaves RTX adhesins and how the receptor DvhD senses c-di-GMP to regulate this process. The paper combines structural biology, biochemistry, and modeling, and the study is technically ambitious. However, the conclusions rely heavily on AlphaFold2 predictions to support structural hypotheses. Given the recent availability and superior accuracy of AlphaFold3, especially for modeling relatively large protein–protein complexes, it is essential that all analyses currently based on AlphaFold2 be repeated or validated using AlphaFold3. This would provide much stronger confidence in the proposed interaction models and substrate recognition mechanisms. In addition, all AlphaFold-based results must be accompanied by confidence metrics such as pLDDT and PAE values to improve transparency and reproducibility.

The Reviewer states that “conclusions rely heavily on AlphaFold2 predictions to support structural hypotheses”. We respectfully disagree with this statement as it seems to misrepresent our study by suggesting that hypotheses remained untested or only supported by structure predictions. As described in our manuscript, in most cases AlphaFold predictions yielded structural hypotheses that we then tested experimentally. Hence, the conclusions of our study rely on the experimental validation of structural models.

Furthermore, we would like to point out that for the sizes of the systems modeled in this study, AlphaFold2 is in principle sufficient. It is also worth pointing out that the accuracy and hence predictive power of AlphaFold2 and AlphaFold3 are comparable, especially for those cases where confidence scores are high (see, for example, <https://doi.org/10.1101/2025.04.16.648930>). Hence, it does not seem justified to disregard models produced by AlphaFold2/ColabFold a priori. In the initial submission, with two exceptions, we only considered AlphaFold2 models with high confidence score (pTM and ipTM > 0.8). Exceptions were some of the models covering the extracellular domains of the adhesins, which received intermediated scores. These adhesin models were only used to illustrate the modular nature of the proteins without considering any mechanistic implications. Another exception is the full-length model for DvhD. In this case, we followed up with experimental structures, confirming key features of the prediction.

Hypotheses directly derived from other models (*i.e.*, protease-substrate interactions) correlate with previous results for the *P. fluorescens* LapG-LapA pair (see doi: 10.1128/JB.00369-15) or were validated in the present manuscript via biochemical and mutational analyses (for DvhG’s structural classification as a BTLCP domain-containing protein and for DvhG-substrate complexes) or via crystal structures (for the domains of DvhD).

In the revised manuscript, we included confidence metrics for all AlphaFold models shown (see Supplementary Table 1). As requested, we also repeated the modeling using AlphaFold3. For monomeric models, there is a good agreement between AlphaFold2 and AlphaFold3 outputs regarding confidence scores and the resulting structural models, assessed by structural alignments. For complexes, AF2 outperformed AF3 in all but one cases, based on ipTM scores and reproducible placement of the substrate peptide at the protease in the five separate models produced by AlphaFold. Since our analysis does not indicate that AlphaFold3 produces models with superior accuracy, we refer in our revised manuscript to the original AlphaFold2-based models that are either comparable or superior to the models obtained with AlphaFold3. Please see Table R1 below for details.

Table R1. AF models and scores

AF models	AF2/Colabfold*			AF3*			AF2 vs AF3
	pLDDT	pTM	ipTM	pLDDT	pTM	ipTM	
DvhG deltaTM	64.9	0.59	-	60.9	0.54	-	0.27**
DvhG BTLCP domain	92.0	0.91	-	94.8	0.93	-	0.25
DvhA 1-108	83.6	0.70	-	91.7	0.82	-	0.79
DvhA 212-483	85.2	0.79	-	72.1	0.63	-	***
DvhA 2224-2824	82.7	0.52	-	83.3	0.53	-	***
DvhA 2832-3038	89.1	0.87	-	88.1	0.87	-	0.34
DVU1545 1-90	77.1	0.62	-	89.7	0.76	-	1.09
DVU1545 144-462	89.8	0.86	-	90.8	0.88	-	0.55
DVU1545 1871-2414	85.9	0.60	-	57.3	0.41	-	****
Pfl _LapG•LapA (88-128)	93.2	0.92	0.87	86.6	0.81	0.55	0.29
Pfl _LapG•MapA (83-131)	93.5	0.92	0.88	88.3	0.82	0.69	0.32
Vc _LapG•CraA (100-141)	92.0	0.91	0.84	90.7	0.86	0.75	0.42
Vc _LapG•FrhA (64-110)	91.3	0.90	0.82	72.4	0.68	0.20	1.00
DvhG (210-399)•DvhA (1-110)	89.7	0.89	0.87	86.8	0.82	0.83	0.70
DvhG (210-399)•DVU1545 (1-91)	89.4	0.89	0.87	88.5	0.82	0.78	1.17
DvhD full-length, dimer	79.7	0.47	0.44	80.4	0.49	0.48	***

* Scores for top-ranked models only. pLDDT: averaged value. Blue, scores > 0.8 (high confidence); grey, scores between 0.6 and 0.8 (intermediate confidence); white, scores < 0.6 (low confidence).

** Long, unstructured N terminus. Alignment using BTLCP domain as reference.

*** High rmsd due to small differences in relative position of domains. Domain-wise rmsd values < 1 Å.

**** AF3 model contains several unstructured regions. Folded domains align with rmsd values < 1 Å.

2. A central weakness of the manuscript is the limited integration between the two parallel stories: the biochemical characterization of DvhG-adhesin interactions and the structural analysis of DvhD. While the paper proposes that DvhD regulates DvhG, no structural or modeling evidence is provided to link these proteins mechanistically. Given the otherwise extensive use of AlphaFold predictions, the absence of attempts to model DvhD-DvhG complexes and of experimental work is conspicuous. I recommend that the paper includes AlphaFold3 modeling of the DvhD–DvhG complex. Without such integration, the manuscript reads as two largely disconnected studies, rather than as a cohesive regulatory model.

As the reviewer may imagine, we have tried extensively to study the DvhD-DvhG interaction, experimentally as well as with modeling using both AlphaFold2 (AF2) and AlphaFold3 (AF3). In previous attempts, the computational models were highly variable with modest to poor confidence scores – see Table R2 below. However, in further extending our modeling attempts to address the reviewer’s comment, we came across one prediction that fulfills criteria for a confident model, *i.e.*, that of a complex of a single dCache domain of DvhD and the BTLCP domain of DvhG modeled with AF3. We now include and describe this model in the revised manuscript (new Fig. 10c). Although the confidence scores for all other DvhD-DvhG complex models based on full-length proteins, isolated domains, and/or different stoichiometries are poor, this type of interaction

seen in the high-confidence model is apparent in some of those other models. Also, we could reproduce this interaction using AF2 with the template option PDB100 (pLDDT=89.4, pTM=0.87, ipTM=0.87). In this case, one of the five models produced a similar interface with high confidence scores (global rmsd of top-scoring AF2 vs AF3 models of 0.96 Å), whereas all five models produced by AlphaFold3 agree with regard to the placement of DvhG on the periplasmic domain of DvhD.

Table R2. AF models and scores for DvhD•DvhG complexes

AF models	AF2/Colabfold ^{*,**}			AF3 [*]		
	pLDDT	pTM	iPTM	pLDDT	pTM	iPTM
DvhD • DvhG	69.0	0.43	0.27	71.8	0.40	0.21
DvhD • DvhG BTLCP	82.4	0.48	0.45	80.9	0.46	0.45
DvhD dCache • DvhG BTLCP	83.3	0.62	0.23	87.8	0.81	0.84
DvhD ² • DvhG	71.1	0.50	0.47	72.1	0.41	0.35 ^{***}
DvhD ² • DvhG ²	-	-	-	67.6	0.38	0.32
DvhD ² • DvhG BTLCP	79.4	0.54	0.53	77.8	0.42	0.39
DvhD ² • DvhG BTLCP ²	78.5	0.45	0.39	76.8	0.39	0.33 ^{***}
DvhD dCache ² • DvhG BTLCP	77.5	0.45	0.21	82.9	0.49	0.26
DvhD dCache ² • DvhG BTLCP ²	84.6	0.81	0.76	74.7	0.37	0.17

^{*} Scores for top-ranked models only. pLDDT: averaged value. Blue, scores > 0.8 (high confidence); grey, scores between 0.6 and 0.8 (intermediate confidence); white, scores < 0.6 (low confidence).
^{**} No templates were used.
^{***} DvhG bound to cytosolic part of DvhD in top-ranking model.

We believe that inclusion of this complex model, as suggested by the reviewer, connects the two parts of our analysis, mitigating the concern raised under this point. Validation of the DvhD-DvhG interaction model depends in part on the ability to observe a direct interaction between the two proteins biochemically, which has not been achieved to date. Hence, we cautiously and explicitly introduce the structure as a model in the Discussion of the revised manuscript. The following text associated with the figure panel was added to the revised manuscript:

(Discussion) “A high-confidence model of a DvhD^{dCache}•DvhG^{BTLCP} complex generated with AlphaFold 3 (AF3) implicates a conserved interface involving helix $\alpha 1'$ of the dCache domain and a surface at the back of DvhG’s active site (Fig. 10c; Supplementary Table 1). The latter surface partially overlaps with the proposed substrate binding path, suggesting that DvhD and substrate binding may be mutually exclusive (Fig. 10d). In addition, the top-scoring structural model shows a disrupted calcium-binding site in DvhG when the protease is engaged with DvhD. These mechanistic hypotheses await experimental validation but present a feasible scenario backed by sequence conservation.”

Furthermore, we see the strength of our manuscript in the comprehensive analysis of the entire system in the context of what is known about the canonical Lap system, which – as a whole – is more impactful than splitting the manuscript into two separate parts.

3. Several other technical issues should also be addressed. The proteolysis assays, while clearly described, would benefit from quantitative analysis of cleavage efficiency using image-based densitometry. Presenting percentages of substrate cleaved, along with basic descriptive statistics (e.g., mean values & standard deviations), would allow more precise comparison across experimental conditions.

As requested, we have quantified the proteolysis assays in the cases where quantitative statements were made. These data are shown in Fig. 4b and 4c of the revised manuscripts. In the revised supplementary material, we include Supplementary Tables 2 and 3 reporting mean values and standard deviations for these quantifications.

4. The crystallographic data are generally acceptable, but the relatively high R-work and R-free values for the periplasmic dCache and HD-GYP domain structures should be explicitly acknowledged. In addition, the deposited coordinates are confusingly labeled: all three PDB entries appear under the same name, “DvhD HD-GYP domain, c-di-GMP-bound,” even though they represent distinct constructs or conditions.

We agree that the low-resolution crystal structures for the dCache and PAS/HD-GYP domain have relatively high R values. However, considering the maximum resolution of 3 Å and 3.44 Å, respectively, these still fall within the range of values reported for structures determined at similar resolution. In the revised manuscript, we now acknowledge explicitly the relatively high R values for the two low-resolution structures. We were careful to only discuss structural features that could be confidently deduced at the resolution at which the structures were determined. The following statement was added:

(Results) “The structures of the periplasmic domain and PAS/HD-GYP fragment of DvhD were determined at relatively low resolution and have R values at the upper end of the spectrum for structures determined at comparable resolutions. As a consequence, we focus our analysis on global features that can be deduced unequivocally from these experimental structures.”

We apologize for the mislabeling of the PDB validation reports. This error was introduced inadvertently during submission of the entries to the PDB. We corrected this mistake. However, the data in the reports was and remains accurate for the three individual structures.

5. The identification of a noncanonical “bridging” c-di-GMP site in DvhD is novel and of clear interest to the field. However, the conclusions about ligand specificity currently rely almost entirely on nanoDSF stabilization assays, which are indirect (No statistical significance was indicated). These data suggest modest stabilization by alternative ligands such as 3’3’-cGAMP, c-di-AMP, and pGpG, but do not establish quantitative binding affinities or stoichiometries. I recommend that the manuscript includes additional binding experiments, such as ITC which can measure K_d , stoichiometries (N), and other thermodynamic parameters. Such data would build on their prior ITC work with c-di-GMP ($N \approx 1.4$), which suggested mixed occupancy of canonical and bridging sites, and would clarify whether alternative ligands engage the same sites or use distinct binding modes. ITC experiments (ideally in 2 or 3 replicates) with targeted site mutants could further discriminate between canonical and bridging site interactions. Most importantly, these additional experiments would address the key question of physiological relevance. At present, it is not clear whether the modest stabilization observed with c-di-AMP, 3’3’-cGAMP, or pGpG reflects biologically meaningful binding in vivo. Quantitative affinity and stoichiometry measurements will be helpful to distinguish these possibilities and help frame a more coherent regulatory model.

We revised the figure showing nanoDSF data by including statistical tests, which support the original statements regarding ligand specificity (Fig. 8b and 8c of the revised manuscript). We also now report ITC data (in triplicates) for the ligands that showed most prominent thermal stabilization using wild-type proteins, *i.e.*, c-di-GMP, cGAMP, pGpG, c-di-AMP (Supplementary Fig. 6D and Supplementary Table 4 of the revised manuscript). These data confirm the

observations from the nanoDSF experiments and additionally indicate a stoichiometry > 1 for the complexes (with the exception of pGpG, which showed poor binding and large errors in individual ITC measurements). These results are consistent with the crystal structure of the HD-GYP domain bound to c-di-GMP showing a 2:3 protein:c-di-GMP complex and previous ITC data reported in Amruta et al., 2024 (doi: 10.1073/pnas.2320410121). The followed text was added to the manuscript:

(Results) “Results are consistent with ITC data that establish c-di-GMP as the ligand with the highest affinity for the PAS/HD-GYP fragment (average effective $K_d=99\pm 7$ nM), followed by 3’3’-cGAMP (average effective $K_d=501\pm 9$ nM) (Supplementary Fig. 6d; Supplementary Table 5)³⁰. Both ligands bind with stoichiometries of >1 (average $N=1.24\pm 0.24$ for c-di-GMP; average $N=1.33\pm 0.36$ for 3’3’-cGAMP), consistent with the structural data that indicated a 2:3 stoichiometry of a HD-GYP•c-di-GMP complex (Fig. 6a). ITC data for pGpG and c-di-AMP confirmed poor binding of these ligands to the PAS/HD-GYP fragment under comparable conditions, associated with large errors (for pGpG) or precluding a quantitative analysis altogether (for c-di-AMP).”

(Discussion) “Based on the structural analysis and c-di-GMP-dependent dimerization of the HD-GYP domain, we suspect that c-di-GMP at the bridging site has a stabilizing role, locking a particular HD-GYP domain dimer conformation in place. Dimerization and overall c-di-GMP binding is sensitive to mutations at both the canonical and bridging site (Fig. 8c; Supplementary Fig. 8), suggesting a cooperative binding event that is coupled with conformational and protein-protein interface changes. Such an interpretation is also consistent with ITC data, that fit best to a single-binding event model, with cooperative c-di-GMP binding resulting in lower K_d values, but potentially masking single-site thermodynamics. In LapD, we did not observe a bridging c-di-GMP binding site in the corresponding c-di-GMP binding EAL domain. Instead, c-di-GMP-induced switching of LapD involves the release of autoinhibitory interactions and the dimerization of its c-di-GMP-bound EAL domains^{16,51}.”

As mentioned in the manuscript, binding of c-di-GMP to the bridging site is dominated by shape complementarity with very few side-chain interactions, which is a challenge for site-directed mutagenesis. In addition, data in Supplementary Figure 8 indicate that dimerization of the HD-GYP domain upon c-di-GMP binding is sensitive to mutations at both the canonical and the bridging site, suggesting a coupling of the sites that in the structure are in close proximity to each other and separated by a flexible loop. These data together with the semi-quantitative nanoDSF data suggest a relevance of both sites for the nucleotide-induced switching of DvhD.

We also expanded the discussion regarding the physiological significance of the other ligands tested. In short, we argue that c-di-GMP is the main physiological ligand, but cannot rule out signaling through 3’3’-cGAMP, which in some deltaproteobacteria could be produced by so call Hypr GGDEF domain-containing proteins. To our knowledge, this SRB lacks enzymes for the production of c-di-AMP, rendering this dinucleotide an unlikely ligand for DvhD. The section reads:

(Discussion) “While c-di-GMP binds to DvhD with high affinity and alters DvhD’s functional state in cells³⁰, we show here that the HD-GYP domain of DvhD, in principle, can also accommodate 3’3’-cGAMP and – with much lower affinity – pGpG and c-di-AMP. To our knowledge, only c-di-GMP signaling systems have been identified in *D. vulgaris* Hildenborough so far. However, one source of 3’3’-cGAMP are hybrid cyclic dinucleotide-producing and promiscuous substrate-binding (Hypr) GGDEF enzymes that are found in other Deltaproteobacteria⁵⁴. Whether 3’3’-cGAMP is physiologically relevant ligand for DvhD in these organisms remains to be established.”

6. The manuscript is focused on description and discussion of the authors' previous work on *P. fluorescens* and *D. vulgaris* with some mention of the *V. cholerae* systems. Given the Lap system is well conserved in Gram-negative bacteria and is of broad interest, the manuscript should include more rigorous discussion and citations on other works in the field such as the Antarctic bacterium *Marinomonas primoryensis*, and slightly different adhesin retention systems such as those in *Salmonella enterica* (e.g. SiiE).

We included references to the mentioned systems in the Introduction, Results, and Discussion sections of the revised manuscript. Of note, our study focuses on processing of adhesins by BTLCP domain-containing proteases and the associated c-di-GMP regulatory system. To our knowledge, there is no LapG homolog identified in *Marinomonas primoryensis* or *Salmonella enterica*. Also, it has not been established so far, that the adhesins in these organisms are regulated by c-di-GMP. Considering these apparent deviations or knowledge gaps, we focus our discussion mainly on organisms with established LapDG systems for adhesin regulation. We see the impact of our study in revealing how a distantly related organism utilizes conserved as well as convergently evolved features to control adhesins, which may inspire work in other organisms.

We added the following text to the manuscript:

(Result) "Such a structure is reminiscent of the membrane-anchoring domain that was first described in a large adhesin from *Marinomonas primoryensis*, and that is a more broadly conserved feature of adhesins that are displayed at the outer membrane^{7,8}."

(Discussion) "Beyond this genus, other bacteria may have evolved systems that are functionally analogous to LapDG and DvhDG for the controlled display of adhesins on the cell surface that await identification. One example is *Salmonella enterica* SiiE, an adhesin that mediates host cell attachment, is associated with a T1SS, and is retained at the outer membrane via its N-terminal moiety⁶⁶. The retention of the adhesin on the bacterial surface is highest during host cell invasion, which may suggest a regulatory step of adhesin release⁶⁷. Whether this step involves proteolytic processing, potentially regulated by cellular signaling, remains to be established."

Other Minor comments to be addressed:

7. There are some inconsistencies in figures and text. Better proofreading will be necessary for the revised manuscript.

For example,

- In Figure 5, the hydrogen bond is attributed to residue E448 in the text, but the figure labels E668.

We apologize for this error, which we now corrected.

- Ensure consistency for "Phosphodiester bond" (e.g. line 309) and "phosphodiesterase bonds" (line 313).

We have corrected the single instance where this mistake was made.

- the phrase following the statement on lines 167-169 describing the second panel of Figure 2B should be revised for better accuracy. "A substrate analog with a di-arginine instead of a di-alanine sequence in the DvhA '[T/A/P]AAG' motif failed to be cleaved in the presence of either protease, confirming the di-alanine sequence as the scissile bond." The figure actually showed relatively faint bands at ~ 26 kDa for all conditions except for the first column (control group with no protease). The PRRG mutation may be more resistant to proteolysis by DvhG and LapG than the PAAG motif.

We modified the statement. It now reads:

(Results) "A substrate analog with a di-arginine instead of a di-alanine sequence in the DvhA '[T/A/P]AAG' motif showed a pronounced resistance to proteolysis in the presence of either protease, confirming the di-alanine sequence as the scissile bond."

- In Fig. 3C, the top panels and bottom panels represent different constructs with the experiments done with the retention module in the bottom panels. Why do the uncleaved proteins in both the top and bottoms appear to have the same size despite the constructs in the bottom panels have the additional retention domain? The full sequence of the constructs used in the proteolysis assay can be included in the supplemental document.

We provide the full sequences in the supplement of the revised manuscript (Table S9). Regarding the apparent size of the constructs, it is important to understand that the samples are not boiled to allow for msfGFP to remain folded. Hence, proteins migrate as partially folded entities, rendering size estimations based on migratory behavior unreliable. We added this explanation to the relevant figure legends:

(Legend to Fig. 4) "Note that samples are not boiled to allow for msfGFP to remain folded. As a result, proteins migrate as partially folded entities, rendering size estimations based on migratory behavior unreliable."

(Legend to Supplementary Fig. 1) "In the unboiled samples, proteins migrate as partially folded entities, retaining the folded state of the msfGFP moiety, rendering size estimations based on migratory behavior unreliable."

- The nucleotide c-di-AMP, which seems to have modest stabilization of DvhD per Figure 7, is known for signalling in bacterial cells, but the biological implication was not discussed. The possible biological significance for the alternative ligands for DvhD, such as linear di-GMP (pGpG), GMP, and GTP could be better discussed.

We expanded the discussion regarding the potential physiological relevance of the alternative ligands. In the original manuscript, we had already mentioned that some HD-GYP proteins use pGpG as a substrate, justifying its inclusions in our study. It is also worth mentioning that the DvhDG system reconstituted in *P. fluorescens* reacts to c-di-GMP, suggesting that this signaling molecule is likely a major and biologically relevant ligand for DvhD. c-di-AMP, which binds poorly to DvhD, may not be a relevant ligand, since the organism lacks the enzymes for its synthesis. The following section describes the rationale (see also Point #5 above for changes to the manuscript text:

“Structural analysis indicated a specific mode of c-di-GMP binding at two distinct sites in the HD-GYP domain of DvhD. Previous studies have shown that enzymatically active HD-GYP domains not only catalyze the first step of c-di-GMP degradation, that is, conversion to linear di-GMP (pGpG), but also, in some cases, the second step to two molecules of GMP^{24,37,46,52,53}. Hence, we cannot rule out pGpG as a ligand for the HD-GYP-like domain of DvhD. Furthermore, some deltaproteobacterial genomes encode enzymatic entities that may produce 3'3'-cGAMP, another bacterial signaling nucleotide with distinct physiological roles⁵⁴. Therefore, given their structural similarity, 3'3'-cGAMP could potentially bind to the same sites as c-di-GMP.”

- Fig. 5C, not useful to superimpose... one can already compare 5B above with 5C

We removed the superposition in panel as requested (Fig. 6c of the revised manuscript).

- Fig. 6C does not explain the dotted versus continuous curves (the light-scattering and refractive index signal) as clearly as Figure S8.

We amended the legend for the corresponding figure (Fig. 7c of the revised manuscript).

- It would be helpful to label temperatures of the peaks. (Fig. 7A and Fig. S6)

We added the additional information to the revised figures (Fig. 8a and Supplementary Fig. 6c of the revised manuscript).

- Statistical analyses, such as ANOVA and t-tests, maybe helpful for nanoDSF results shown in Figure 7.

We included statistical analyses as requested (Fig. 8b and 8c of the revised manuscript).

- The gel Figure S1C should show the top 3 bands on the protein marker. The size of these markers supports the size of the bands given asterisks (** and *** in particular). This would strengthen claim that the fluorescent bands have twice the size of the monomers and indicate possible dimerization.

We included the marker labels for the top three bands in the revised figure. However, proteins migrate as partially folded entities since samples were not boiled to retain the folded state of the msfGFP moiety, rendering size estimations based on migratory behavior unreliable. A corresponding statement was added to the figure legend. See also above.

Reviewer #2 (Remarks to the Author):

Thank you for your time and the constructive review. We appreciate your efforts.

Reviewer #3 (Remarks to the Author):

In essence, this manuscript describes a comparative study of one system with another that has been more extensively investigated. It focuses on the deltaproteobacterial Dvh system, comparing it with the gammaproteobacterial Lap system. In their cognate hosts, both systems are responsive to the second messenger c-di-GMP to regulate biofilm formation and bacterial dispersal. Their components are functionally analogous but structurally distinct. In both systems, a protease processes adhesins which are critical for cell attachment. The results in this manuscript extended the functional analogy of Dvh to Lap at the mechanistic/biochemical level and explored the implications of their structural divergence in maintaining the functionality of the former.

The authors proposed a model wherein the Dvh system functions similarly as the Lap system. When c-di-GMP concentration is high, the transmembrane DvhD (LapD counterpart) binds c-di-GMP through its cytoplasmic portion. This binary complex sequesters the protease DvhG (LapG equivalent, periplasmic), preventing it from cleaving and freeing the adhesins from the cell surface. Lack of proteolytic processing allows the adhesins, DvhE and DVU1545 (LapA equivalents), to be anchored to the cell surface through a retention (R) domain located in the periplasm. In this state, these adhesins are attached to the cell surface to facilitate and promote biofilm formation. When c-di-GMP concentration is low, the apo-DvhD releases DvhG, which then cleaves off the R domain of the adhesins. The anchorless adhesins slip out of cells to allow bacterial dispersal from biofilms.

The authors took a multiple disciplinary approach in this study. There are two parts to this manuscript, both of which used AlphaFold modeling to demonstrate the structural differences and similarities of Dvh components with their Lap counterparts. In the first part, they used *E. coli*, a gammaproteobacterium, to express proteases of both systems. Fractionation was used to investigate protein localization and whole cell lysates expressing the proteases for proteolytic processing of their respective substrates. Although these experiments are not perfectly clean, appropriate controls were included to ensure the conclusions are solid. The second part of the manuscript focused on DvhD, the transmembrane c-di-GMP receptor of the Dvh system. They purified three different parts of DvhD: 1) its periplasmic dCache-like domain, 2) its entire cytoplasmic part (PAS + HD-GYP domains) and 3) the HD-GYP domain by itself. The last two truncated variants were shown to bind c-di-GMP with the longer one being a dimer and the shorter one being a monomer that dimerizes in the presence of c-di-GMP. Moreover, they determined the X-ray crystal structures of these three with the surprising discovery of a third c-di-GMP molecule binding at the interface of the HD-GYP dimer.

The overall quality of the work and the manuscript is high, with substantial new experimental results and insights. The proposed model is consistent with the authors' observations and appears highly plausible. One major issue is that the authors have attempted to tackle nearly all aspects of their model, some of which are more strongly substantiated than others.

Thank you for the thorough assessment of our study. We see the main impact of our work in providing a system-wide comparison. At the same time, we are aware of the points that remained unanswered or are supported less than others, which we addressed transparently in our revised manuscript.

Major points:

1. There is no direct evidence or information for the periplasmic interaction between DvhD and DvhG and the effect of c-di-GMP on this interaction. Bacterial two-hybrid system may be considered, especially because c-di-GMP levels in *E. coli* could be artificially manipulated.

We thank the reviewer for the suggestion. Indeed, as mentioned in the manuscript, we have no experimental evidence for a direct interaction between DvhD and DvhG. This is not because of lack of trying, using both biochemical (*i.e.*, pull downs using the isolated domains or full-length proteins; bacterial two-hybrid system (BACTH)-based approaches) and computational approaches (*i.e.*, modeling with AF2 and AF3). Although we can purify DvhD and DvhG as full-length proteins in detergent, we did not observe any sign of co-purification, independent of the addition of c-di-GMP. Also, DvhG is prone to degradation and aggregation over time. Similar experiments using purified domains, which have worked for the LapDG system, yielded negative results for DvhDG.

For the bacterial two-hybrid (BACTH) assays, we initially observed positive signals for the full-length DvhDG pair. While encouraging at first, testing DvhD with an unrelated membrane protein, PilJ from *P. aeruginosa*, also produced a positive signal, indicating a false-positive result and rendering the initial observation unreliable. We next used the BACTH system with the isolated periplasmic domain of DvhD and the BTLCP domain of DvhG. In this experiment, we also included a DvhD periplasmic domain attached to a leucine zipper motif to mimic a dimeric receptor. While the BACTH positive controls produced a robust signal, the results for the domains of DvhD and DvhG were negative for any combination tested. In light of the inconclusive or negative results across multiple approaches, we chose an honest discussion and hope we can address this point in the future, *e.g.* by looking for yet-to-be-discovered factors that could aid in stabilizing a specific DvhDG complex.

In the meantime, we extended interaction modeling using AF2 and AF3. See also our response to Reviewer 1 (points #1 and #2) for detail. This approach yielded one model job with high overall confidence scores, including ipTM (scoring the protein-protein interaction interface). All five individual models show nearly identical binding poses, suggesting a robust result. Interestingly, this interaction is observed sporadically in modeling attempts involving full-length proteins (or full-length DvhD with the BTLCP domain of DvhG). Given the poor confidence scores of the latter, we only show the domain-wise, high-confidence interaction model in the revised manuscript (new Fig. 10). Due to the technical challenges mentioned above, experimental validation remains beyond the scope of the current manuscript.

2. In the context of c-di-GMP binding and its effect on DvhD, is there an experimental structure of PAS/HD-GYP in complex with c-di-GMP? Was there an attempt to soak the PAS/HD-GYP crystal with c-di-GMP or crystallization in the presence of c-di-GMP? This seems logical as the PAS/HD-GYP fragment clearly binds c-di-GMP from the current and past results. It would be better to compare the apo form of PAS/HD-GYP with its c-di-GMP complex than with the binary complex of HD-GYP and c-di-GMP.

To our knowledge, there is no experimental structure of a PAS/HD-GYP module in complex with c-di-GMP. The closest is the structure of the GAF/HD-GYP module of *PmGH* that we refer to in our manuscript (Supplementary Figure 6B in the revised submission).

We would have loved to compare apo and nucleotide-bound crystallographic states of the PAS/HD-GYP construct. Consistent with the conformational change c-di-GMP binding likely induces in DvhD as based on the HD-GYP•c-di-GMP structure, soaking has not yielded a corresponding structure. We extensively screened for crystallization conditions using our purified DvhD PAS/HD-GYP protein in the presence of c-di-GMP without any success so far.

3. In light of the unexpected c-di-GMP binding at the interface in the crystal structure of the HD-GYP domain, it would be helpful to revisit the binding of c-di-GMP to this domain and to the PAS/HD-GYP fragment directly with more analytic techniques (than thermostability). More nuanced experimental design and data analysis may reveal multiple binding affinities and more precise stoichiometry. In this context, the authors' previous publication in PNAS showed that one molecule of PAS/HD-GYP binds 1.4 molecules of c-di-GMP, consistent with 3 ligands for one dimer. Related to point #2) above, the biophysical data (ITC and thermostability) suggest possibly a higher resolution for the crystal structure of PAS/HD-GYP if complexed with c-di-GMP.

As mentioned under the previous point, we have tried extensively to obtain a ligand-bound PAS/HD-GYP protein structure without success. While we see an increase in thermostability in the presence of c-di-GMP, this conformation appears to resist crystallization. This is not uncommon, especially considering the major impact c-di-GMP is likely to have on the overall conformation of DvhD's PAS/HD-GYP domains.

In the revised manuscript, we expanded the analysis of nucleotide binding to the PAS/HD-GYP protein via ITC (see also our response to Reviewer 1, Point #5). We could reproduce earlier results that consistently indicate a stoichiometry >1 for c-di-GMP and high-affinity binding (Supplementary Figure 6D; Supplementary Table 4). The data did not indicate multiple, independent binding sites and data fits were optimal when inferring a "one set of sites" binding model. One reason for this observation could be that localized binding goes along with protein conformational changes, with the breaking and forming of protein-protein interfaces convoluting the analysis. Another reason could be a cooperative binding event. Our analysis of wild-type and mutant HD-GYP proteins suggests a coupling of the two c-di-GMP binding sites, consistent with an apparent single binding event.

As described in our manuscript, the PAS/HD-GYP fragment is a constitutive dimer, whereas the isolated HD-GYP domain is monomeric in the absence of c-di-GMP but dimerizes when c-di-GMP is added to the solution. We utilize this difference in our experimental design. In the SEC-MALS experiments, we use dimerization upon c-di-GMP binding as an analytical readout. ITC experiments were carried out with the PAS/HD-GYP fragment, which is dimeric with and without c-di-GMP, removing dimerization from the equation (although we still suspect protein-protein interfaces at the dimer interface to change upon binding based on the available structural data). Of note, ITC data using the isolated HD-GYP domain of DvhG were uninterpretable under comparable conditions used for the experiments with the PAS/HD-GYP protein. We suspect that agitation, prolonged incubation times required for ITC titrations, and monomer-dimer transitions are complicating factors in this experiment. Regarding the latter, note that we only observed dimerization of the HD-GYP domain when c-di-GMP was added to the mobile phase of the gel filtration step. This result suggests that the dimer is unstable when the ligand is removed during the gel filtration. We observed a similar phenomenon with the cytoplasmic domain of LapD, although the structural basis is likely to be different given the differences in domain organization between LapD and DvhD.

Please see our response to Reviewer 1 under Point #5 for specific changes to the text that address this point.

4. There is too much focus on the Lap system in the writing. I suggest to focus more on Dvh with Lap for comparison and interpretation purposes after presenting results. A couple of examples below.

Thank you for this assessment. Our study is motivated by the comparison of the two systems, and it seems appropriate and equally effective to first describe what we know for one system, which motivates the experiments we describe in the present manuscript on the other system. Since the data are not changed by the order or emphasis, we feel that this stylistic choice should not alter the impact of our study.

a. The first two paragraphs under the subheading “Adhesion retention domain...” could be switched. Consider using the Lap system for comparison and data interpretation instead of a lead-in for the Dvh experiments.

The work on the Lap systems motivates the experiments we conducted on the DvhDG system. It seems appropriate to describe this context in the order we chose. Also, as described in this section, the modeling of the well-established LapG-substrate pairs provides the proof of concept for the analysis of DvhG and its substrates. We feel it is important to establish first the validity of an approach before applying it to less established systems.

b. Sentence starting on L242 would be a better opening for the relevant paragraph. The remaining part of the paragraph could be re-structure to state something like: “Unlike LapD, which has xxx domain for c-di-GMP binding, DvhD contains an HD-GYP domain with the potential to bind c-di-GMP. We investigated.....”. In this context, the structure of the periplasmic domain seems out of place as its proposed interaction with DvhG is not investigated in anyway.

We have reworded this section following the Reviewer’s suggestion:

(Results) “***DvhD contains a periplasmic dCache domain.*** A recent study has shown that DvhD enables c-di-GMP-dependent control over DvhG in a *P. fluorescens* strain lacking LapD and LapG, supporting that DvhD could be the functional homolog of LapD in SRB³⁰. However, DvhD bears no sequence or domain resemblance to LapD (Fig. 5a). Unlike LapD, which has a cytosolic S-helix/GGDEF/EAL domain module responsible for c-di-GMP binding, DvhD contains an HD-GYP domain with the potential to bind or cleave c-di-GMP^{16,31}. In LapD, c-di-GMP binding to the cytosolic domain generates a conformational change that is transduced to its periplasmic LapD/MoxY domain^{2,16,36}.”

The last point is somewhat mitigated by Fig. 10 of the revised manuscript, which presents a plausible model for a DvhDG complex based on AlphaFold modeling and sequence conservation (see our response to Reviewer 1, Point #2).

5. It seems that the biochemical/biophysical studies of DvhD should precede the structural studies in the organization of the manuscript. These include c-di-GMP binding and analysis of oligomeric states.

This is a stylistic choice. Several of the biochemical/biophysical studies serve as direct validation of the structural studies. Hence, we introduced the structural models first, which indicated novel features that required validation. To follow the logic of the way the experiments were motivated (and performed), we chose to keep the original order.

Minor points:

6. More informative summary paragraphs or concluding sentences would be helpful.

Almost every Results section ends in a summary statement highlighting the main outcome.

7. There should be explicit indications of *D. vulgaris* as a deltaproteobacterium and all others are gammas in the writing. Speculate a little on whether the cases here represent structural divergence or functional convergence in evolution.

We addressed this point in the revised text. We addressed the speculative part in the last section of the Discussion in our original manuscript:

(Discussion) “While DvhD bears a radically different architecture to LapD, other components of the Dvh system (DvhG, DvhA, and T1SS) are similar to their counterparts in the Lap system, both in sequence conservation and function^{25,29,30}. Therefore, it seems unlikely that the entire system is a case of convergent evolution. One possibility is that there was a horizontal gene transfer event from some organism that had the Lap system, but that the LapD component was not transferred or was lost. DvhD could have evolved from a HD-GYP domain-containing phosphodiesterase, either host-resident or exogenously acquired, to replace LapD.”

8. It is unclear why *E. coli* is used here instead of *P. fluorescens*, which allowed a successful reconstitution of the Dvh system for functional analysis.

Ideally, we would have liked to reconstitute the system entirely with purified proteins. Unfortunately, DvhG is unstable in its pure form, which required working in lysates as mentioned in the original version of the manuscript. *E. coli* provides higher expression levels and is commonly used for protein expression. Also, *E. coli* has inherently low c-di-GMP levels and a far less complex c-di-GMP signaling network than *P. fluorescens*, minimizing complicating factors.

Reviewer #4 (Remarks to the Author):

The manuscript by Font and colleagues describes a structural and biochemical study of the DvhD/G/A system, responsible for the synthesis and exposition of adhesins in *Desulfovibrio vulgaris* Hildenborough. This system is present in Sulfate-reducing bacteria and relate to the LapG/D/A system that exists in Gammaproteobacteria.

The authors have assemble a remarkable number of evidence, based on structural modelling, crystal structures, clever biochemical assays and mutagenesis to illuminate the mechanism of

regulation of adhesin fixation and release at the outer membrane of *Desulfovibrio vulgaris* Hildenborough. In particular, this work provides a detailed mechanism and a structural basis of the cleavage activity and activation of DvhG as well as of ci-diGMP sensing by the DvhD membrane protein.

I have found the evidence compelling, the structural work excellent except for one crystallographic structure that could be improved (see below) and the model strongly supported by the data.

Thank you for the positive feedback and your time reviewing our manuscript.

1. One putative drawback is that no experiment has been done *in vivo* to see if the mechanism described exist in the *Desulfovibrio vulgaris* Hildenborough. It would have been nice if some mutants had been introduced *in vivo* to inspect the consequences on biofilm formation in this bacteria or alternatively in the model system *Pseudomonas fluorescens*.

Our study builds on previous work that demonstrated reconstitution of the system in *P. fluorescens* (see <https://doi.org/10.1073/pnas.2320410121>). Some of the new insight from our structural and biochemical work cannot be tested at the moment in this system, especially those requiring the retention domain of the *D. vulgaris* adhesins, as constructs containing this domain are non-functional in *P. fluorescens*. Published studies on the effect of gene deletions targeting the adhesins and associated secretion system established their central role in biofilm formation of the native organism (doi: 10.1128/jb.00379-24; doi: 10.1128/mbio.01696-17). Given the extent of our current study and the fact that we currently do not have the growth and genetic manipulations of this organism established in our lab, we feel an *in vivo* characterization of the system in *D. vulgaris* is beyond the scope of the current work.

2. Another issue is the structure of the DvhD PAS/HD-GYP domain solved at 3.4Å (Table S1). There seem to be 1% Ramachandran outlier which is rare. Moreover, the R-free and R-factor are quite high as well, with 0.33/0.31 suggesting that more work can be done on the model. Can the author provide a model and structure factor of this structure so that we can verify that this is a real outlier? Alternatively, a bit more work on the structure refinement might help so that we can be sure that the model is fully refined.

We acknowledge the high R values for the structure, which are within but at the upper end of the range for structures determined at similar resolution. However, we would like to point out that we were careful in our analysis and the claims we made based on the structure to avoid overinterpretation. The crystallographic data for the PAS/HD-GYP fragment is not only of low resolution but also contains indications of ice rings. Despite these drawbacks, the crystal reported on here yielded the best data set by far; attempts to reach higher resolution or optimal freezing remained unsuccessful to this date. Also, in response to Reviewer 1 (Point #4), we included an explicit statement regarding the refinement statistics and level of structural analysis in the revised manuscript. The section reads as follows:

(Results) “The structures of the periplasmic domain and PAS/HD-GYP fragment of DvhD were determined at relatively low resolution and have R values at the upper end of the spectrum for structures determined at comparable resolutions. As a consequence, we focus our analysis on global features that can be deduced unequivocally from these experimental structures.”

In some parts, in particular the introduction, the paper is written for specialist of these systems and thus, except for a few aficionados, it might not be comprehensible for some non experts of the field. I have tried to include some examples below. Many of the domain names are not defined. Including these definition would really help. Moreover a clear definition of the representation of this system in the various bacteria would be a plus. The supplementary dataset S1 (presented in discussion) does not really help. Can the author provide a simple bacterial phylogenetic tree with the order/genus that encode for these systems? OR something more comprehensible?

We have included domain definition and other clarifying text in the revised manuscript. Also, Fig. 9b of the revised manuscript now shows a phylogenetic distribution of DvhDG at the class level based on a conservative census. We kept the more comprehensive analysis in Supplementary Dataset 1 as it provides a broader view of the genetic context of *dvhDG*.

Otherwise I have few minor comments in order to improve an already very good manuscript.

Details of the minor issues:

L101 102 what are the PAS and HD-GYP domains ? Are these really important for understanding the paper ? if yes then they should be defined. What does PAS or HD-GYP stand for ?

We have expanded the Introduction to address this point. The section now reads:

(Introduction) “While these results support that DvhG and DvhD perform functions that parallel those of the *P. fluorescens*-native Lap system, it is interesting to note that DvhD and DvhG have a sequence and domain organization distinct from those of their functional counterparts in the Lap system. Specifically, DvhD contains a hitherto undefined periplasmic domain as well as cytoplasmic Per-Arnt-Sim (PAS) and HD-GYP domains^{31,32}. PAS domains occur in all kingdoms of life and often possess sensory function, whereas HD-GYP domains, named after their conserved sequence motifs, bind and, in many cases, cleave cyclic dinucleotides. The DvhD domain organization contrasts with LapD’s LapD/MoxY periplasmic domain followed by a cytoplasmic, catalytically inactive HAMP-GGDEF-EAL module. Notably, GGDEF and EAL domains often function as diguanylate cyclases and c-di-GMP-specific phosphodiesterases, respectively, but in some cases also provide sensory functions such as in the case of LapD³³. How DvhD senses c-di-GMP levels and transduces this signal to regulate adhesion and biofilm formation in *D. vulgaris* remains poorly understood.”

L124 L141 « correlating with the lack of a lapD-like gene » I thought both the drawing (Fig1) and introduction suggest that DvhD is an homologue of LapD ? (for instance L98 ?). A clearer explanation arrives only L244 with « A recent study has shown that DvhD, enables c-di-GMP-dependent control over DvhG in a *P. fluorescens* strain lacking LapD and LapG, supporting that DvhD could be the functional homolog of LapD in SRB 25 . However, DvhD bears no sequence or domain resemblance to LapD, leaving the open question of how DvhD senses c-di-GMP levels and transduces this signal to regulate adhesion and biofilm formation in *D. vulgaris* ». Can the author consider reformulating these parts so that it is clearer from the introduction ?

In the introduction, we state that DvhD and DvhG are functional homologs to LapD and LapG with distinct differences in their domain organization. We agree that the section in L141 (original manuscript) requires clarification that matches the introduction. See our response to the previous point for the amended text. Also:

(Results) “This pocket appears to be occluded in DvhG, correlating with the lack of sequence similarity between the periplasmic domains of LapD and DvhD³⁰.”

L149. « comprises a predicted (targeting) helix with low sequence identity, » is this supposed to mean low sequence conservation ?

We introduced the wording suggested by the Reviewer.

L271; dCache. Please define what dCache domains are and what they are used for in bacteria. Are these specific to bacteria?

We added the following sentence to this section to address the point above:

(Results) “The bilobed dCache domains are part of the Cache family comprising predominantly extracytosolic ligand binding domains found in bacteria, archaea, and eukaryotes³⁹ .”

Figure 2 would have been easier to follow with a schematic description of the adhesin domains and boundaries

We added a domain diagram for both adhesins in Fig. 2a of the revised manuscript.

Figure 6 juxtamembrane should be juxtamembrane

We fixed the spelling mistake in Fig. 7a of the revised manuscript. Thank you for pointing it out to us.

Reviewer #5 (Remarks to the Author):

This manuscript by Font et al. focuses on molecular mechanism of adhesion/dispersion and biofilm formation in a sulfate-reducing bacterium. Structural analysis including three X-ray crystal structures and Alpha Fold2 prediction and detailed biochemical analysis provide a deep insight about the mechanism involving key proteins, DvhD, DvhG, and DvhA. It is interesting that the signal by c-di-GMP binding to DvhD is transduced from cytosol to periplasm in outward direction over the inner membrane in the Dvh systems. It is expected that the comprehensive study attracts wide attention of not only specialists of the related fields but also the general readership of *Nature Communications*. Potentially, this work is valuable to be published in this journal. However, the explanation or description was insufficient in some parts, and several important points should be reconsidered before publication.

Thank you for your constructive review.

Major concerns

1. Lines 454

The titrated site number ($n = 1.4$ per protein mol) calculated from ITC measurement in a previous study (ref. 25) is supportive for Author conclusion that 3 mol of c-di-GMP is bound to HD-GYP domain. K_d value, another parameter obtained from the ITC experiment, was single value (322

nM, ref. 25). This means that the three potential binding sites for c-di-GMP have nearly identical affinity and that single transition for the c-di-GMP binding is observed in the ITC experiment. However, I guess that among three potential binding sites observed in X-ray crystal structure of the complex, two canonical sites for c-di-GMP should have clearly higher affinity than that of a single bridging site, because a dense network of hydrogen bonds and electrostatic interactions are formed only for the canonical sites. If that is the case, two titration steps would be measurable. Is this inconsistent with Author conclusion? According to Fig. 5D, the number and degree of conservation of the interacting amino acid residues at the bridging site (#, 3 conserved/4 positions) are less than those at the canonical sites (§, 7 conserved/10 positions). Alternatively, I suspect that the c-di-GMP binding at the bridging site is low affinity site and an artifact under crystallization conditions in which 1 mM of high concentration of c-di-GMP (3000 folds of K_d) is contained.

The ITC experiment reported in our previous manuscript (doi: 10.1073/pnas.2320410121) and repeated in the present manuscript is consistent with a high affinity binding event. Given that the bridging site involves only very few side-chain interactions and is dominated by shape complementarity, sequence conservation may not be a reliable measure to assess binding strength or for comparison with other proteins. At the same time, we do not claim that the bridging site is a conserved feature within the HD-GYP protein family. Also, our mutational analysis suggests coupling of the bridging and canonical sites, which are in close proximity to each other. In responds to Reviewer 1, we repeated the ITC measurements, obtaining consistent results to our previous findings with stoichiometries >1 . Data fit best to a 'one set of sites' model. We infer that we observe a cooperative binding event. In general, cooperative binding at interfaces lowers the apparent K_d , but may mask individual site thermodynamics. Data analysis may be further complicated by large-scale conformational changes of the protein upon c-di-GMP binding, likely involving changes in intra- or intermolecular protein-protein binding interfaces. Hence, our measurements only yield apparent or effective parameters. See also our response to point #2 for the amended Discussion text.

See our response to Reviewer 1, Point #5 for text changes to the Results and Discussion sections.

2. When nevertheless Authors claim that one more c-di-GMP is really bound at the bridging site in the sulfate-reducing bacteria, Authors need to discuss physiological significance of the binding of three molecules of c-di-GMP in the DvhD signal sensing mechanism in comparison with LapD dimer in which the binding of two molecules of c-di-GMP is required.

Likely, the third c-di-GMP-binding site at the DvhD dimers serves a stabilizing role. It is an observation based on the crystal structure and SEC-MALS data that we discuss in our manuscript. Determining binding stoichiometries in cells will not be possible, hence our discussion is based on the biochemical/biophysical and structural data presented here. We suspect that the analysis may be complicated by cooperative binding of c-di-GMP to both canonical and bridging sites and/or conformational changes involving changes in protein-protein interfaces. We extended the discussion as follows:

(Discussion) "Based on the structural analysis and c-di-GMP-dependent dimerization of the HD-GYP domain, we suspect that c-di-GMP at the bridging site has a stabilizing role, locking a particular HD-GYP domain dimer conformation in place. Dimerization and overall c-di-GMP binding is sensitive to mutations at both the canonical and bridging site (Fig. 8c; Supplementary Fig. 8), suggesting a cooperative binding event that is coupled with conformational and protein-protein interface changes. Such an interpretation is also consistent with ITC data, that fit best to

a single-binding event model, with cooperative c-di-GMP binding resulting in lower K_d values, but potentially masking single-site thermodynamics. In LapD, we did not observe a bridging c-di-GMP binding site in the corresponding c-di-GMP binding EAL domain. Instead, c-di-GMP-induced switching of LapD involves the release of autoinhibitory interactions and the dimerization of its c-di-GMP-bound EAL domains^{16,51}.

While c-di-GMP binds to DvhD with high affinity and alters DvhD's functional state in cells³⁰, we show here that the HD-GYP domain of DvhD, in principle, can also accommodate 3'3'-cGAMP and – with much lower affinity – pGpG and c-di-AMP. To our knowledge, only c-di-GMP signaling systems have been identified in *D. vulgaris* Hildenborough so far. However, one source of 3'3'-cGAMP are hybrid cyclic dinucleotide-producing and promiscuous substrate-binding (Hypr) GGDEF enzymes that are found in other Deltaproteobacteria⁵⁴. Whether 3'3'-cGAMP is physiologically relevant ligand for DvhD in these organisms remains to be established.

Crystal structures suggest that DvhD's HD-GYP domain undergoes a conformational change upon binding c-di-GMP, splaying the stalk-like helices connecting the PAS and HD-GYP domains. We propose that, similar to LapD, this conformational change extends through the transmembrane helices and periplasmic domain of DvhD, ultimately controlling DvhG activity³⁰.

In addition, this study showed that DvhG is anchored on the inner membrane. Does this feature also have any physiological meaning in the Dvh system?

As we discussed in our manuscript original manuscript already, transmembrane localization is not a conserved feature of DvhG orthologs. Hence, we do not believe this is an essential feature.

3. Figure 6A, Lines 339-340

This study suggests that the large conformational change of stalk helices of DvhD dimer is induced upon the binding of c-di GMP by comparing N-terminal position of the stalk helices between free PAS/HD-GYP domain and c-di GMP-bound HD-GYP domain. However, the conformation change upon the ligand binding is estimated using two constructs having different domain architectures. Is it OK? It is likely that the PAS region of the free PAS/HD-GYP domain interact each other in the homodimer. Therefore, lack of PAS domain may affect the difference of conformation in c-di-GMP-bound HD-GYP domain. If the conformational change of the stalk helices is really induced by c-di-GMP binding, the remaining N-terminal parts including dCACHE domain, trans-membrane helices, and PAS domain will be separated from the counter part?

The lack of equivalent structures for different states is a caveat of many studies using isolated domains or domain combinations. Since we were not successful so far in obtaining alternative crystal structures (e.g., apo-HD-GYP or c-di-GMP-bound PAS/HD-GYP), we were careful in not overinterpreting our results, acknowledging this limitation. As stated in the manuscript, one prominent difference is the disorder of a loop that bridges the canonical and bridging site; the loop contains binding residues that when mutated in HD-GYP affect dimerization and stabilization; further suggesting a cooperative event. Based on our analysis, we do think that c-di-GMP will alter dimeric interfaces. As stated in the original manuscript, dimerization involves mainly the helical stalks with little to now contributions by the PAS or HD-GYP domains. One could envision scissor motions or twisting, changing the relative orientation of the domains. We discuss this aspect in light of the model for a DvhDG complex in the Discussion (see above our response to Point #2).

In addition, in crystal lattice, the N-terminal region of c-di-GMP-bound HD-GYP domain may be also affected by the adjacent molecules. Does the crystal packing affect the conformation of stalk helices? These points should be mentioned in the manuscript. Although X-ray crystal structure of DvhD PAS/HD-GYP domain is low resolution, the structures of the binding site of c-di-GMP and the surrounding residues in the c-di-GMP-bound HD-GYP domain will provide the information for structural change upon the c-di-GMP binding, expecting the details of initial trigger for structural change. Did Authors check it?

Crystal contacts could have an impact on protein conformation. However, these interactions are typically considered weak. Although we cannot rule out at the moment that crystal packing induced the observed conformation, it seems equally if not more plausible, that crystallization trapped a native conformation the protein adopts in solution, considering the presumed strength of the interactions. We include the following statement in the revised manuscript, which is based on an interface analysis:

(Results) “Although crystallization may have an influence on protein conformation, crystal packing interfaces are rather weak (calculated Δ^iG values of -0.9 to -4.8 kcal/mol for interfaces involving the stalks; compared to dimer interfaces with Δ^iG values of -21.3 and -24.7 kcal/mol)⁵⁰ .”

Considering the low resolution of the PAS/HD-GYP fragment crystal structure and high R values associated with the refinement, we opted to forgo a more detailed analysis of the c-di-GMP-induced conformational changes. However, we described the flexibility of a loop in the apo state that bridges the canonical and bridging site, a major structural difference between the two states.

Minor concerns

1. To clearly express distinct molecular architectures between DvhD and LapD, LapD structure should be also shown in the manuscript. The overall dimer model of LapD predicted by Alpha Fold2 is useful for the comparison with DvhD. Authors are able to add the comparison to Figure 4 or as new supplemental data.

Since LapD and DvhD contain completely different domains that can be distinguished bioinformatically (shown in Fig. 5a), an AlphaFold model of a LapD dimer may not add much value here. Also, it would not accurately depict its solution conformation that we have described in our previous work (see <https://doi.org/10.7554/eLife.21848>). For DvhD, we lack comparable data and hence resort to the AlphaFold model for illustrating a possible dimer, which we validated by determining crystal structures of the individual domains of DvhD. Also, we need to balance comments from other Reviewers who pointed out the emphasis on LapDG that was already perceived as (too) extensive.

2. Fig. 4C legend

A sentence “Dashed and solid curves show the light-scattering and refractive index signal from a size exclusion-coupled ...” means “Dashed and solid curves show the light-scattering and refractive index signals, respectively, from a size exclusion-coupled ...”?

For easier understanding SEC-MALS data for most of readers, Authors also cite an appropriate reference for the SEC-MALS experiments here.

We amended the figure legends and added a reference to a general SEC-MALS description to the specific Material and Methods section, as requested.

3. Several residues that are not found in Figures were cited in the main text: E448 (line 307; E668?), Y562A (line 396; Y652A?), etc. If they are mistakes for residue numbers, please fix them. Or the positions of those residues were not shown in Figures. The positions of the cited residues must be indicated in the corresponding figures. In addition, thermal stability data about M646D was not found in Figs. 7B and 7C (line 397). Please check again over the entire manuscript.

These were indeed mistakes that have been corrected in the revised text. We also removed references to mutation M⁶⁴⁶D in the revised manuscript.

4. Many figures and tables lack explanation for contents in their legends. Authors should provide self-explanatory legends. For example, Table S2 has no legend. To understand this table, most of readers require explanation of many abbreviations contained in the table: “Pred.” “SP”, “GLOB”, “S”, and “O”. Reference for DeepTMHMM should be also cited here. Authors should again check whether all figures and tables in the main manuscript and extended data have self-explanatory legends and whether abbreviations are defined here or elsewhere.

We added legends to the Tables S6 and S7. References to the bioinformatics tools can be found in the appropriate sections of main manuscript that refer to the Tables S6 and S7 as well as Table S8 (Resource Table).

5. Line 493

b-D-1-493 thiogalactopyranoside - β-D-1-493 thiogalactopyranoside (change to symbol font)

We made the requested change.